# The radiative feedback continuum from Snowball Earth to an ice-free hothouse

**Ian Eisenman** [1] ✉ **& Kyle C. Armour** [2]

Paleoclimate records have been used to estimate the modern equilibrium climate sensitivity. However, this requires understanding how the feedbacks governing the climate response vary with the climate itself. Here we warm and cool a state-of-the-art climate model to simulate a continuum of climates ranging from a nearly ice-covered Snowball Earth to a nearly ice-free hothouse. We find that the pre-industrial (PI) climate is near a stability optimum: warming leads to a less-stable (more-sensitive) climate, as does cooling of more than 2K. Physically interpreting the results, we find that the decrease in stability for climates colder than the PI occurs mainly due to the albedo and lapse-rate feedbacks, and the decrease in stability for warmer climates occurs mainly due to the cloud feedback. These results imply that paleoclimate records provide a stronger constraint than has been calculated in previous studies, suggesting a reduction in the uncertainty range of the climate sensitivity.

Recent community assessments[1,2] have substantially narrowed the estimated range of Earth's equilibrium climate sensitivity (ECS) for the first time in decades, leading to better constraints on future warming[3]. This narrowing of the uncertainty in the ECS (which is defined as the equilibrium global-mean surface temperature response to $CO_2$ doubling from pre-industrial levels) was achieved in large part through the use of paleoclimate records from times when the climate was substantially different from today. In Sherwood et al.[1], the ECS likelihoods derived from proxy reconstructions of temperatures and estimates of radiative forcing during the Last Glacial Maximum (LGM) and mid-Pliocene warm period (mPWP) provided the strongest line of evidence against high ECS values. In another assessment[2], proxy reconstructions of LGM, mPWP, and Eocene temperatures also informed the strongest line of evidence against high ECS values: so-called "emergent constraints" wherein a relationship between temperature changes and ECS within an ensemble of Earth System Models (ESMs) is combined with observations or paleoproxy reconstructions of those temperature changes to derive a constraint on ECS.

A confounding factor in the use of paleoclimate records to inform the sensitivity of the modern climate to greenhouse gas forcing is that the radiative feedbacks governing the climate response can vary with the underlying climate itself[1,2]. That is, the use of paleoclimate records to constrain ECS requires understanding how modern radiative feedbacks (which govern ECS) relate to radiative feedbacks operating in climates much colder or much warmer than today.

Following previous work[4–6], Sherwood et al.[1] represented the dependence of radiative feedbacks on the underlying climate by including a quadratic feedback term in the standard model of global energy balance used to relate reconstructions of temperature and climate forcing to modern-day ECS. This approach typically represents the net radiative feedback as becoming less negative (i.e., a more-sensitive climate) with global warming and more negative with global cooling (e.g., ref. 1). While higher-order terms that are cubic and beyond in surface temperature could be included, they are typically assumed to be small and omitted. This raises key questions regarding the range of temperatures over which this approximation applies, what causes it to fail outside this range, and relatedly how confident we can be in the structure of the radiative feedback dependence on global temperature over a wide range of climate states. The answers to these questions also have implications for emergent constraints, in which the mapping of feedbacks between past and future climate states is implicitly accounted for through the use of ESMs to simulate the paleoclimate states and ECS values on which the constraints rely.

Here we warm and cool a state-of-the-art ESM to simulate a continuum of climates ranging from a nearly ice-covered Snowball Earth to a nearly ice-free hothouse planet. We analyze how the radiative

---

[1]Scripps Institution of Oceanography, University of California San Diego, La Jolla, USA. [2]Department of Atmospheric Sciences and School of Oceanography, University of Washington, Seattle, USA. ✉e-mail: eisenman@ucsd.edu

feedbacks depend on the underlying climate, and we physically interpret the results.

## Results and discussion
### Climate model simulations

Using NCAR's Community Earth System Model Version 2 (CESM2)[7] in its standard workhorse configuration, we ramp $CO_2$ concentrations over a range of 11.5 doublings. Specifically, we start from the end of a 500-year pre-industrial (PI) control simulation, which has a constant $CO_2$ concentration of 284.7 ppm, and we either increase or decrease the atmospheric $CO_2$ concentration at a rate of 1% per year (Fig. 1a). The Warming simulation, which extends a preexisting gradual $CO_2$ quadrupling simulation[8], is 279 years long and ends with a $CO_2$ concentration of 4522 ppm, which is 16 times the PI value. The Cooling simulation is 514 years long and ends with a $CO_2$ concentration of 1.6 ppm, which is 1/175 times the PI value. See the Methods for details.

This leads to a 59K range in simulated annual-mean global-mean surface temperature, with climates ranging from a nearly ice-covered Snowball Earth to a nearly ice-free hothouse planet. Averaged over the last decade of the PI control simulation, the global-mean surface temperature is 15 °C, and the ice area is 11.4% of the global surface area. The latter includes sea ice, snow cover on land, and prescribed time-invariant glacial ice cover (see the Methods for details), with sea ice covering 6.1% of the ocean (4.3% of the globe). In the Warming simulation, the annual-mean global-mean surface temperature increases by 18K to 33 °C (Fig. 1b), and the annual-mean ice area decreases to 3.2% of the globe (Fig. 1c), with sea ice covering 0.0% of the ocean. In the

Cooling simulation, the temperature decreases by 41K to −26°C (Fig. 1b), and the ice area increases to 68.7% of the globe (Fig. 1c), with sea ice covering 70.3% of the ocean.

The surface temperature in the deep tropics (averaged annually and over 10°S–10°N) is 28 °C in the PI (Fig. 1e), and it reaches 42 °C in the final decade of the Warming run (Fig. 1f) and 5 °C in the final decade of the Cooling run (Fig. 1d). In the PI climate, the polar surface temperature (averaged annually and over both hemispheres poleward of 70°) is −24 °C, and this region is largely covered with snow and ice (Fig. 1h). In the final decade of the Warming run, the polar temperature reaches 4 °C (Fig. 1f), and the remaining ice cover is almost exclusively glacial ice (Fig. 1i), which is a specified surface type in CESM2 with an area that does not evolve during the simulations. In the final decade of the Cooling run, the polar temperature reaches −68 °C (Fig. 1d), and the ice cover extends into the tropics (Fig. 1g).

### Net radiative feedback and effective climate sensitivity

In order to evaluate the net radiative feedback over this continuum of climates, we adopt the standard model of global energy balance and climate feedbacks:

$$\Delta N = \Delta F_{GHG} + \Delta F_{net} = \Delta F_{GHG} + \lambda_{net}\, \Delta T, \qquad (1)$$

with

$$\lambda_{net} \equiv \frac{\Delta F_{net}}{\Delta T} = \frac{\Delta N - \Delta F_{GHG}}{\Delta T}. \qquad (2)$$

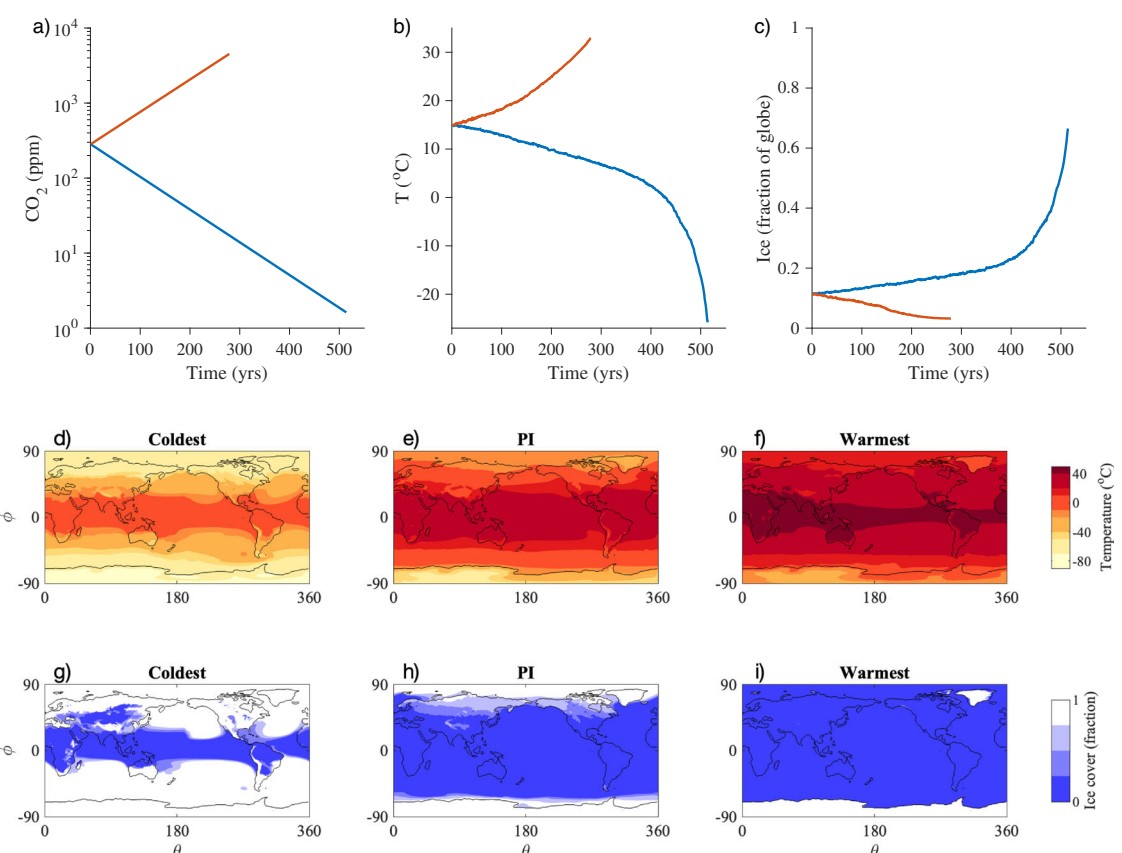

**Fig. 1 | Forcing and climate response in CESM2 simulations.** Time series of (**a**) specified atmospheric $CO_2$ volume mixing ratio, (**b**) annual-mean global-mean surface temperature $T$, and (**c**) annual-mean global ice area (including sea ice, snow cover on land, and glacial ice), in the Warming simulation (red) and the Cooling simulation (blue). Also included are surface temperature maps averaged over the last decade of the (**d**) Cooling, (**e**) pre-industrial (PI) control, and (**f**) Warming simulations, as well as ice area maps averaged over the last decade of the (**g**) Cooling, (**h**) PI control, and (**i**) Warming simulations (with latitude $\phi$ and longitude $\theta$). Note that we use the relatively short averaging period of a single decade in these maps in order to better capture the full range given the rates of change near the end of the Warming and Cooling simulations.

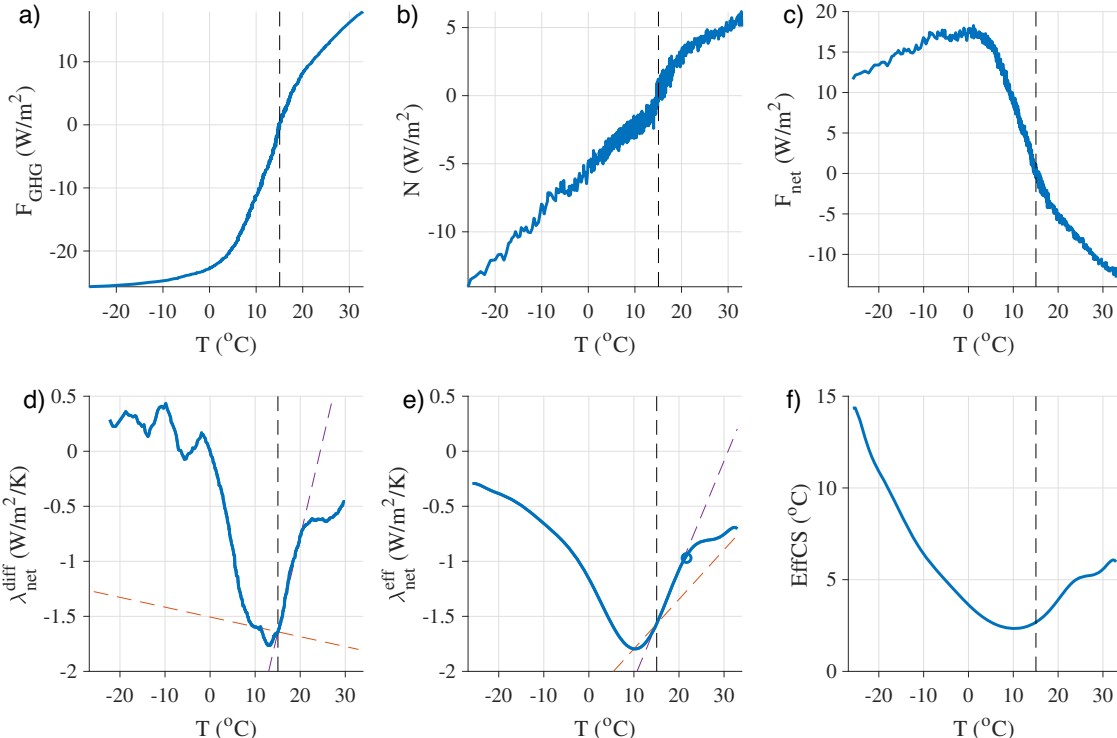

**Fig. 2 | Dependence of the net feedback and effective climate sensitivity on the underlying climate. a** $CO_2$ radiative forcing $F_{GHG}$. **b** Top-of-atmosphere (TOA) net energy flux $N$. **c** Net radiative response of the climate system, $F_{net} \equiv N - F_{GHG}$. **d** Net differential feedback parameter $\lambda_{net}^{diff}$. **e** Net effective feedback parameter $\lambda_{net}^{eff}$. The blue circle indicates the result from a previous analysis of an instantaneous $CO_2$ quadrupling simulation with the same climate model[51]. **f** The effective climate sensitivity EffCS. All quantities are plotted versus the annual-mean global-mean surface temperature $T$. The dashed lines in **d**, **e** indicate a linear dependence of $\lambda_{net}$ on $\Delta T$ that runs through the pre-industrial (PI) climate and either a climate 5K colder (red) or a climate 5K warmer (magenta). In all panels, the vertical dashed line indicates the PI climate.

Here all quantities are averaged annually and globally: $N$ is the top-of-atmosphere (TOA) net energy flux reported by the model (using top-of-model fields), $F_{GHG}$ is an estimate of $CO_2$ radiative forcing relative to PI based on previously published line-by-line radiative transfer calculations[9] (see Methods and SI Fig. S1), $F_{net} \equiv N - F_{GHG}$ is the net radiative response of the climate system, $\lambda_{net}$ is the net radiative feedback parameter, and $T$ is the surface temperature. The fluxes are defined to be positive in the downward direction, and the feedback parameter is negative for a stable climate. The modifier $\Delta$ is described below.

The radiative forcing $F_{GHG}$ and resulting value of $N$ are plotted in Fig. 2a, b, and the difference $F_{net}$ is plotted in Fig. 2c. It can be readily seen that $F_{net}$ does not depend linearly on $T$. Specifically, the slope of the $F_{net}$ versus $T$ curve (Fig. 2c) is most negative near the PI climate (black vertical dashed line), being less steep in warmer and colder simulated climates. In extremely cold climates, the slope becomes zero around $T = 0\,°C$ and then changes sign for climates with $T < 0\,°C$, implying that additional incremental levels of cooling lead to less energy coming into the climate system and hence more cooling.

We consider two approaches to define $\lambda_{net}$ in Eq. (2), following previous work[10] (see Methods for details):
(i) The "effective feedback" $\lambda_{net}^{eff}$, which describes the radiative feedback processes operating between a given climate state and the PI climate. In this case, we define $\Delta$ as the anomaly from the PI climate, and Eq. (2) is calculated from $F_{net}$ after applying a polynomial smoothing. Note that this allows $\lambda_{net}^{eff}$ to vary smoothly even in the limit $\Delta T \to 0$ (see Methods and SI Fig. S2).
(ii) The "differential feedback" $\lambda_{net}^{diff}$, which describes the feedback processes operating within a given climate. Hence $\lambda_{net}^{diff}$ is the local tangent value of the slope in Fig. 2a. In this case, we define $\Delta$ as the

anomaly associated with an incremental change in climate, and Eq. (2) is calculated using a regression of $F_{net}$ versus $T$ within a running window (see Methods).

The effective feedback may be seen as most directly relevant to current discussions of ECS, since they often involve estimates of past climates compared with today, rather than estimates of past climate variability (e.g., refs. 1,2). On the other hand, the differential feedback reflects the radiative response to a temperature perturbation in a given underlying climate, and hence it may be somewhat easier to physically interpret.

The net feedback parameter calculated using each of these approaches is plotted in Fig. 2d, e. A striking result is that the PI climate is near the stability optimum. The differential feedback $\lambda_{net}^{diff}$, which indicates the stability of the climate system to perturbations, is most negative when the global temperature is 2K cooler than the PI value (Fig. 2d). Starting from the PI, warming leads to less-stabilizing radiative feedbacks and hence a more-sensitive climate, as does cooling of more than 2K. The effective feedback $\lambda_{net}^{eff}$ shows similar behavior, being most negative when the global temperature is 5K cooler than the PI value (Fig. 2e).

The TOA net energy flux when the climate has reached equilibrium is $N = 0$, as is approximately the case in the simulated PI climate (Fig. 2b). Hence from Eq. (2), the equilibrium warming response to a change in $CO_2$ is $\Delta T = -\Delta F_{GHG}/\lambda_{net}$. This is known as the ECS in the special case of a doubling of $CO_2$ from PI levels, as mentioned above. It is given by ECS $= -F_{2\times}/\lambda_{2\times}$, where $F_{2\times} = 4.2$ W/m$^2$ is the value of the radiative forcing $\Delta F_{GHG}$ when $CO_2$ is doubled from its PI value of 284.7 ppm, and $\lambda_{2\times}$ is the value of the feedback parameter $\lambda_{net}^{eff}$ operating in this climate state. For other climate states, the effective climate sensitivity (EffCS) is similarly defined using the effective feedback

parameter:

$$\text{EffCS} \equiv \frac{-F_{2\times}}{\lambda_{net}^{\text{eff}}}. \qquad (3)$$

The EffCS is plotted in Fig. 2f. This shows that the sensitivity is lowest near the PI climate, with more-sensitive climates at warmer and much colder temperatures. The continuum of simulated climates spans a range of EffCS values from 2 °C to 15 °C. Note that the EffCS (Fig. 2f) scales as the inverse of $\lambda_{net}^{\text{eff}}$ (Fig. 2e).

Under more-extreme cooling, the value of $\lambda_{net}^{\text{diff}}$ in Fig. 2d becomes positive when the global temperature drops below approximately 0 °C, which is 15K colder than the PI climate. At this point there is a change in the sign of the slope of the $F_{net}$ versus $T$ curve in Fig. 2c: as the temperature drops below this point, incremental coolings are accompanied by incremental decreases in the level of heating by the net radiative response of the climate system. This corresponds to the Snowball Earth bifurcation point, beyond which the sea ice in the model expands toward the equator in an irreversible process. Note that this is the point at which the global temperature and ice area begin to abruptly change in the Cooling simulation (Fig. 1b, c). The implications of this change in the sign of $\lambda_{net}^{\text{diff}}$ can be illustrated using a simple single-layer model of the climate system, which is described in SI Section S1. The positive value of $\lambda_{net}^{\text{diff}}$ implies that the climate is transitioning across a range of temperatures for which the only equilibrium climate state is unstable (SI Fig. S3). Previous studies have demonstrated that bifurcations and bistability associated with the Snowball Earth climate occur in climate models of varying levels of complexity in certain ranges of $CO_2$ and solar luminosity[11–15]. Note that $\lambda_{net}^{\text{eff}}$ remains negative for all climates, in contrast with $\lambda_{net}^{\text{diff}}$, which illustrates how the EffCS and $\lambda_{net}^{\text{eff}}$ framework can give potentially misleading results about the stability of the underlying climate state because it is based on anomalies from the PI climate.

Under warming, the values of $\lambda_{net}^{\text{eff}}$ and $\lambda_{net}^{\text{diff}}$ increase monotonically. Notably, the climate remains stable ($\lambda_{net}^{\text{diff}}$ is negative) even at extreme levels of global warming nearing 15K above the PI. Note that previous studies using idealized single-column radiative models have found that the net climate feedback becomes more negative with warming for climates warmer than approximately 25K above the PI[16,17].

Note that when the climate is forced to transiently evolve away from an equilibrated state, it is possible for the climate feedback parameter to become less negative due to the spatial pattern of surface temperature changes[18]. In SI Section S1, we investigate the extent to which this may explain the results in Fig. 2d, e by using a standard two-layer model of the climate system[19] that includes a term to represent the deep ocean heat uptake efficacy. The results show that although deep ocean heat uptake efficacy can cause $\lambda_{net}^{\text{diff}}$ and $\lambda_{net}^{\text{eff}}$ to become less negative under both warming and cooling as the climate gets farther from its equilibrated state, a moderate (CMIP5-mean) deep ocean heat uptake efficacy leads to far smaller changes in $\lambda_{net}$ than we find in CESM2 (SI Fig. S4). Furthermore, even with a large ocean heat uptake efficacy, the two-layer model results in a "V"-shaped feedback dependence on temperature that is centered at the equilibrated climate (purple curve in SI Fig. S4), in contrast to the "U" shape centered at a temperature several degrees colder than the PI that we find in CESM2 (Fig. 2d, e). This suggests that the changes in $\lambda_{net}$ shown in Fig. 2d, e are considerably outside of what would be expected from changing surface temperature patterns associated with deep ocean heat uptake, and that feedback nonlinearities with global temperature are the main cause of the dependence of the net feedback on the underlying climate in CESM2 over the simulated range considered here.

## Linear representation of $\lambda_{net}(T)$

Many recent studies have suggested that colder climates are more stable than warmer climates, including climates considerably colder than the PI. Specifically, as summarized in a previous assessment[2], paleoclimate records[20–28] and comprehensive climate models[29–36] suggest a general trend toward less-stabilizing radiative feedbacks (hence higher EffCS) with increasing global temperature over a range of climates spanning approximately 6K colder than today to approximately 10K warmer than today. However, the results presented here suggest that the PI climate is near a stability optimum, with warming and cooling beyond 2K both leading to less-stable climates (Fig. 2d). Similarly, warming and substantial cooling both lead to less-negative values of $\lambda_{net}^{\text{eff}}$ and higher EffCS (Fig. 2e, f). While the climate at the temperature characteristic of the LGM (4-6K colder than the PI) is more stable than the simulated climates that are warmer than the PI, consistent with the studies mentioned above, we find that climates beyond about 6K colder than the PI can be considerably less stable than climates warmer than the PI.

As noted in the Introduction, previous work has typically represented nonlinearities in the dependence of the net radiative response on the underlying climate by using a quadratic relationship with global temperature (e.g., ref. 1). In this case, Eq. (1) is replaced with

$$\Delta N = \Delta F_{GHG} + \lambda_0 \, \Delta T + \frac{1}{2} \, \alpha \, \Delta T^2, \qquad (4)$$

where $\lambda_0$ is the net feedback near the PI climate and $\alpha$ is a coefficient scaling the nonlinear radiative response. This implies a linear dependence on global temperature for both the effective feedback and the differential feedback:

$$\lambda_{net}^{\text{eff}} = \lambda_0 + \frac{1}{2} \, \alpha \, \Delta T \text{ and } \lambda_{net}^{\text{diff}} = \lambda_0 + \alpha \, \Delta T. \qquad (5)$$

Note that here we adopt the formalism used in Sherwood et al.[1].

Sherwood et al.[1] use the value $\alpha = 0.1$ W/m$^2$/K$^2$ (with an uncertainty of ± 0.1 W/m$^2$/K$^2$) for the difference in the feedback at the LGM compared with the PI, and they implicitly assume no change in feedback between the PI and warmer climates. We include red dashed lines in Fig. 2d, e to represent a linear dependence of $\lambda_{net}$ on $T$ that goes through the PI climate ($T = 15$ °C) and the climate with an LGM-like level of cooling ($T = 10$ °C). The slopes of the curves correspond to values of $\alpha = -0.01$ W/m$^2$/K$^2$ for $\lambda_{net}^{\text{diff}}$ and $\alpha = 0.05$ W/m$^2$/K$^2$ for $\lambda_{net}^{\text{eff}}$. We also include for comparison magenta dashed lines that go through the PI climate and the climate at 5K of warming ($T = 20°C$), which have slopes that correspond to values of $\alpha = 0.17$ W/m$^2$/K$^2$ for $\lambda_{net}^{\text{diff}}$ and $\alpha = 0.10$ W/m$^2$/K$^2$ for $\lambda_{net}^{\text{eff}}$. Note that CESM2 has previously been shown to be among the ESMs with the largest values of $\alpha$ when assessed for temperature changes near the PI climate[6].

These results show that the value of $\alpha$ adopted by Sherwood et al.[1] for the change in $\lambda_{net}$ at the LGM is much larger than in the CESM2 results, because the "U" shape in Fig. 2d, e causes the feedback at 5K of cooling to be similar to the feedback at the PI. This suggests that paleoclimate records provide a stronger constraint on the upper bound of the ECS than has been calculated in previous estimates such as Sherwood et al.[1]. In other words, if we were to repeat the analysis of Sherwood et al.[1] using the value of $\alpha$ that we find here for the difference between the LGM and PI feedbacks, our lower value of $\alpha$ would imply a lower modern-day climate sensitivity than they found. This amounts to a stronger constraint on the upper bound of the ECS than they report and therefore a reduction in the uncertainty range for the ECS.

This is because the CESM2 results suggest that the LGM may be a more direct analogue to current warming than previously assumed, since the feedbacks are relatively similar. In other words, Sherwood et al.[1] took $\lambda_{net}$ to be more negative at the LGM than the modern value, whereas we find that the feedbacks are similar. So a given paleo estimate of the LGM value of $\lambda_{net}$ implies a similar modern feedback value according to our results, whereas the analysis of Sherwood et al.[1]

would take it to imply a less-negative modern feedback and hence a more-sensitive modern climate.

For climates warmer than the PI, these results imply a larger value of $\alpha$ that is actually somewhat similar to the Sherwood et al.[1] result. But here the value of $\alpha$ applies to warming rather than cooling. Overall, this suggests that feedback nonlinearities could be large for future warming, consistent with some other studies (e.g., ref. [6]), while being relatively small for colder climates similar to the LGM.

These results suggest that the formulation of feedbacks as changing linearly with global temperature applies only over a narrow range of climates, and that because of the "U" shape of the relationship between the feedback and the underlying climate, comparing feedbacks between two climates depends sensitively on the temperatures of the climates.

The range of climates over which the quadratic term in the global energy budget (Eq. (4)) serves as a useful approximation can be seen by comparing the linear fits to the values of $\lambda_{net}^{diff}$ and $\lambda_{net}^{eff}$ (magenta lines in Fig. 2d, e). For the effective feedback $\lambda_{net}^{eff}$, the quadratic term captures much of the variation in the feedback parameter for climates with $T$ between about 3K colder and 8K warmer than the PI climate, thus serving as a decent approximation to feedback changes over a temperature range spanning the PI climate and $CO_2$ quadrupling but not spanning climates as cold as the LGM. Outside of this temperature range, the quadratic approximation fails spectacularly. For the differential feedback $\lambda_{net}^{diff}$, the quadratic term provides a decent approximation over a similar temperature range. Including additional terms in the Taylor series expansion (i.e., order $\Delta T^3$ and higher in Eq. (4)) would be expected to widen the range over which the expansion provides a useful approximation.

## Individual radiative feedback parameters

In order to identify what physical processes are responsible for the decrease in stability under both cooling and warming from the PI (Fig. 2b), we begin by using a radiative kernel analysis to assess which individual feedback parameters are driving the changes. We use radiative kernels[37] that were generated based on CAM5, which is the previous version of the atmospheric model in CESM2. Using these kernels, we compute the annual-mean global-mean change in the radiative response associated with changes in the surface temperature ($F_P$ for Planck feedback), atmospheric lapse rate ($F_L$ for lapse-rate feedback), humidity ($F_w$ for water-vapor feedback), and surface albedo ($F_\alpha$ for albedo feedback). We compute the cloud radiative response ($F_c$) as the difference between the sum of the individual feedbacks and $F_{net}$; hence $F_c$ also includes the residual ($F_{res}$) due to inaccuracies in the radiative kernel analysis (see SI Section S2 for details). Each of the resulting radiative responses is shown in SI Fig. S5. Note that the radiative kernel analysis effectively linearizes the simulated response to changing climate fields about a climate near the PI. Although it would be more accurate to use radiative kernels that vary with the climate[38], the present analysis could be seen as a preliminary step toward building understanding of feedbacks across a wide continuum of climate changes by using a kernel that does not vary with climate, before considering how the radiative kernels change.

We define each individual feedback parameter as

$$\lambda_i \equiv \frac{\Delta F_i}{\Delta T}, \qquad (6)$$

where the subscript $i$ can indicate any individual feedback and $\Delta$ has the same two definitions as in Eq. (2).

The results (Fig. 3) indicate that the decrease in stability (i.e., $\lambda_{net}^{diff}$ becoming less negative) for climates more than 2K colder than the PI is caused by the lapse-rate and albedo feedbacks, whereas the decrease in stability for climates warmer than the PI is caused mainly by the cloud feedback. The roles of these feedbacks occur robustly in both

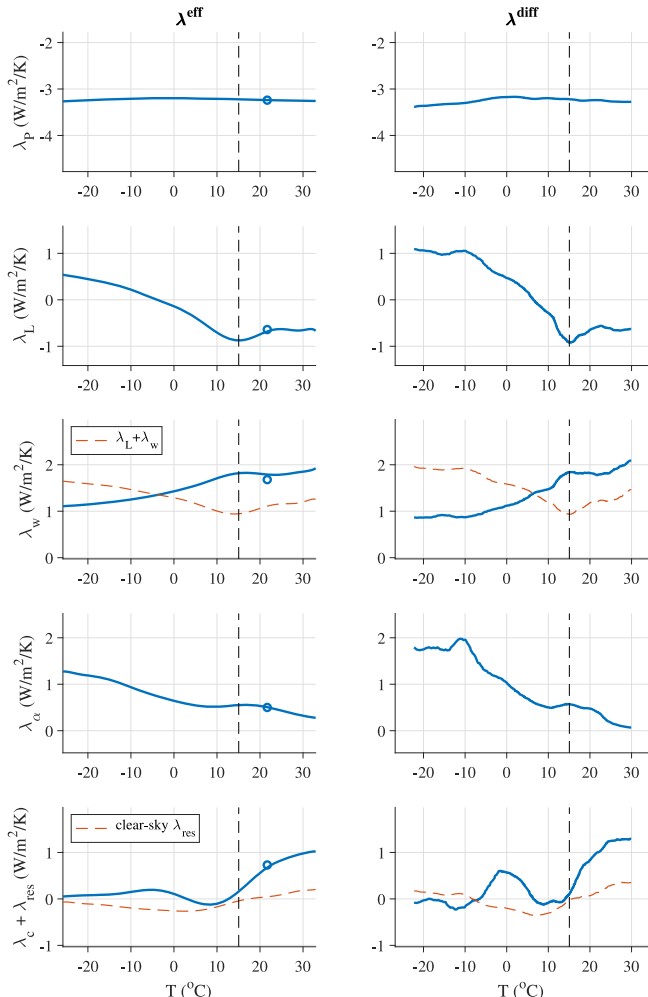

**Fig. 3 | Individual feedback parameters computed using radiative kernels:** Planck $\lambda_P$, lapse rate $\lambda_L$, water-vapor $\lambda_w$, albedo $\lambda_\alpha$, cloud $\lambda_c$, and the residual term $\lambda_{res}$. The effective feedback parameters $\lambda^{eff}$ are plotted in the left column, and the differential feedback parameters $\lambda^{diff}$ are plotted in the right column. Panels have different vertical ranges but the same vertical scale for comparison. The sum of the lapse-rate and water-vapor feedbacks is also indicated in the third row, and the clear-sky result for the residual is also indicated in the fifth row. The vertical dashed line in each panel indicates the pre-industrial (PI) climate. The blue circles indicate the results from a previous analysis of an instantaneous $CO_2$ quadrupling simulation with the same climate model[51] for comparison.

the differential feedback analysis (right column in Fig. 3) and the effective feedback analysis (left column in Fig. 3). Note that although there is some compensation between the lapse-rate feedback ($\lambda_L$) and the water-vapor feedback ($\lambda_w$), as expected, the changes in the combined feedback ($\lambda_L + \lambda_w$) are dominated by the lapse-rate feedback (see red dashed lines in third row of Fig. 3).

## Physical interpretation of feedback changes

Here we interpret the results in Fig. 3. We focus on the differential feedback parameters, since they describe the physics of a given climate and hence may be more-readily understood than the effective feedback parameters.

The large range of simulated climate changes may be expected to be annually and zonally uniform to a first approximation. Hence we repeat the analysis in Fig. 3 taking the annual average and the zonal average of each kernel as well as each simulated climate field before multiplying the kernels by the climate fields (see SI Section S3 for details). We find that the result matches closely with the feedback

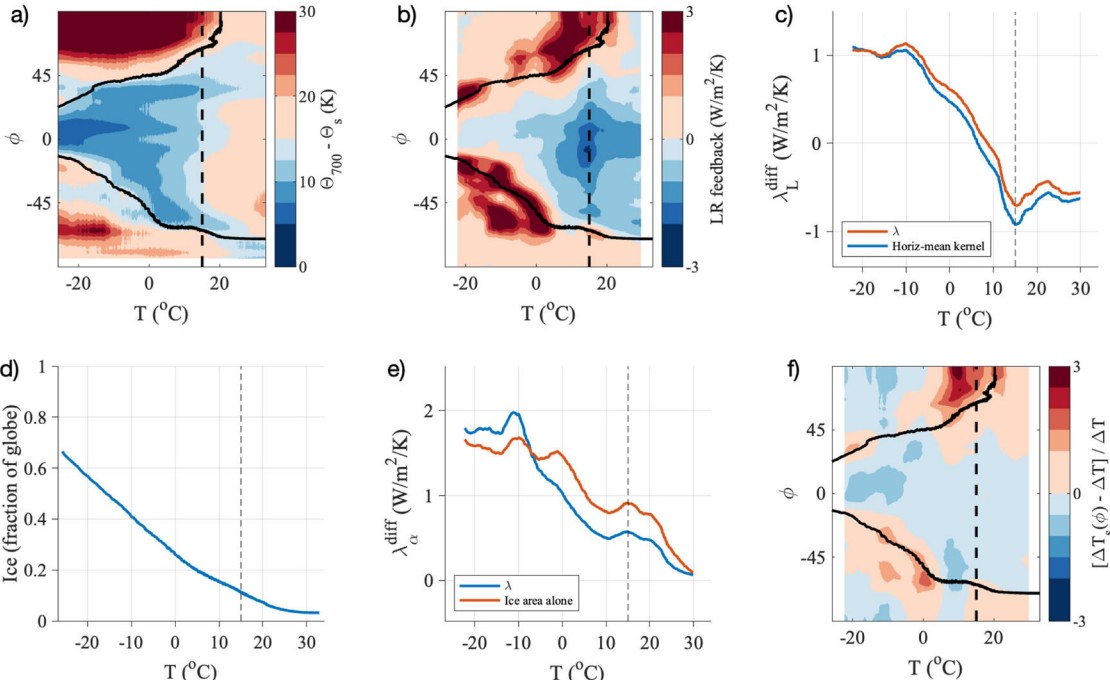

**Fig. 4 | Physical interpretation of changes in individual feedback parameters.**
**a** Inversion strength, plotted as the difference in annual-mean zonal-mean potential temperature $\Theta$ between the 700-hPa level ($\Theta_{700}$) and the surface ($\Theta_s$). **b** Spatial structure of the annual-mean zonal-mean lapse-rate feedback parameter value (see SI Section S3). The spatial mean of this field gives the differential lapse-rate feedback parameter as estimated using annual-mean global-mean fields (SI Fig. S6). **c** Lapse-rate feedback parameter $\lambda_L^{\mathrm{diff}}$. The red line is an approximation using only the global-mean atmospheric temperature profile, given by Eq. (S19) in SI Section S3, and the blue line is $\lambda_L^{\mathrm{diff}}$ repeated from Fig. 3. **d** Global ice area (as in Fig. 1c).

**e** Albedo feedback parameter. The red line is an approximation using only the sensitivity of the total ice area to global temperature (i.e., the slope of the curve in **d**), given by Eq. (S21) in SI Section S3, and the blue line is $\lambda_\alpha^{\mathrm{diff}}$ repeated from Fig. 3. **f** Pattern of amplified surface warming, shown as the change in the local departure of the zonal-mean temperature from the global-mean temperature, normalized by the change in the global-mean temperature (see SI Section S3). The black vertical dashed line in each panel indicates the pre-industrial (PI) climate. In **a**, **b**, and **f**, the black solid line indicates the 50% contour of the ice cover. All quantities are plotted versus the global surface temperature $T$.

parameters computed using the full 4-dimensional structure of the simulated climate and kernel fields (SI Fig. S6). This suggests that the zonal and seasonal patterns of temperature, surface albedo, humidity, and cloud changes do not play a substantial role in the variations in each feedback parameter shown in Fig. 3, allowing the specific factors driving the variations in each feedback parameter to be more-readily assessed by examining only the meridional and vertical structure of the fields.

The decrease in stability with cooling in cold climates is the main novel result of the present study, since previous work has discussed the decrease in stability with warming. Hence we begin by interpreting the lapse-rate and albedo feedbacks.

**Lapse-rate feedback.** The lapse-rate feedback describes the impact of changes in the vertical temperature structure. In the tropics today, deep convection occurs, and the temperature profile is close to being moist adiabatic. Warming causes the moist adiabatic lapse rate to decline. This is a negative local feedback, since it means that smaller changes in surface temperature are needed to bring about a given change in outgoing longwave radiation. On the other hand, in the present-day Arctic the planetary boundary layer is often capped by a temperature inversion and hence a very stable stratification, which suppresses vertical mixing and causes temperature changes at the surface not to be propagated aloft, which is a positive local feedback.

The inversion strength can be described by the difference in potential temperature between the 700-hPa level and the surface (cf. ref. 39), which is plotted in Fig. 4a. Across the range of simulated climates, ice-covered regions of the globe tend to have an inversion, as expected because the surface absorbs less solar radiation when it is

covered with snow or ice, setting up the potential for a positive lapse-rate feedback in these regions. This leads to a less-negative global lapse-rate feedback as the climate cools and more of the globe resembles the present-day Arctic (Fig. 4a, b).

As the climate warms and sea ice is lost, the erosion of polar inversions leads to less-positive polar lapse-rate feedbacks (Fig. 4a, b). However, the lapse-rate feedback in the tropical region becomes less negative with warming for climates warmer than the PI. An analysis of a previous version of this model lead to fairly similar changes in the spatial pattern of the lapse-rate feedback parameter under varied levels of forced warming[40]. The mechanisms driving the changes in the tropical temperature profile that cause this are beyond the scope of the current study. The result is that the global lapse-rate feedback becomes somewhat less negative with warming, at least up to temperatures about 10 K warmer than the PI climate, although the cloud feedback dominates the changes in the net feedback parameter for climates warmer than the PI.

The temperature feedback radiative kernel has a spatial structure that varies vertically but is fairly uniform horizontally, suggesting that lapse-rate feedback changes should approximately track changes in the vertical structure of globally-averaged atmospheric warming. Indeed, we get a similar result when we repeat the analysis using the global-mean temperature profile (red line in Fig. 4c), which removes the influence of horizontal variations in the efficiency of radiation to space but still retains vertical variations. This result implies that the changes in the lapse-rate feedback parameter under cooling global temperature are dictated primarily by the global-mean atmospheric temperature profile becoming more similar to the Arctic today, causing the global lapse-rate feedback to approach the positive value in the Arctic today.

**Albedo feedback**. The albedo feedback occurs because a warmer climate has less ice cover, and ice-free regions absorb more solar radiation rather than reflecting it back to space. We find that the albedo feedback increases approximately monotonically with cooling global temperature across the range of simulated climates. The albedo feedback radiative kernel has a spatial structure with values most negative in the low latitudes, where there is the most incident solar radiation. Nonetheless, we find that the migration of the ice edge into sunnier latitudes has a relatively limited influence on the variations in the albedo feedback parameter: we get a fairly similar result when we repeat the analysis using a spatially-uniform radiative kernel, which removes the influence of spatial variations in incident solar radiation as well as clouds and other factors (red line in Fig. 4e). In this case the albedo feedback parameter is approximated to be proportional to the sensitivity of the ice area to global temperature (i.e., the slope in Fig. 4d). This implies that the albedo feedback becomes more destabilizing primarily because the ice area expands more rapidly with cooling in colder climates.

This behavior continues in climates warmer than the PI, with the change in ice area per change in global temperature continuing to decrease as the climate warms (Fig. 4d), leading to a smaller albedo feedback in warmer climates (Fig. 4e). In the warmest simulated climates there is almost no remaining snow and sea ice (Fig. 1i), and the albedo feedback $\lambda_\alpha^{\text{diff}}$ approaches zero (Fig. 3).

**Cloud feedback**. Clouds cause shortwave cooling and longwave heating, and changes in clouds with climate lead to a feedback that can be either positive or negative. We find that the cloud feedback in CESM2 is approximately zero near the PI climate, but the feedback becomes increasingly destabilizing as the underlying climate warms. Previous work using CESM2 and earlier versions of this model similarly found that cloud feedbacks are more destabilizing in warmer climates[29,36,41].

An important caveat associated with the changes in the cloud feedback shown in Fig. 3 is that this term includes the residual due to factors including inaccuracies in the radiative kernel analysis. One measure of this is the residual when the kernel analysis is repeated using clear-sky fields (see "Caveats" section below), which we find contributes about 25% of the diagnosed cloud feedback change between the PI and warmest simulated climates (red dashed line in bottom right panel of Fig. 3).

We also carry out an alternative test of the impact of clouds that does not rely on the radiative kernels. Instead, we redo the net feedback analysis in Fig. 2 using clear-sky fields reported by the model for the change in TOA net energy flux $\Delta N$. The resulting values of $\lambda_{net}^{\text{eff}}$ and $\lambda_{net}^{\text{diff}}$ are plotted in SI Fig. S8. For both measures of the net feedback in SI Fig. S8, the feedback remains relatively constant in climates warmer than the PI when using clear-sky fields, whereas it becomes steadily less negative with warming when using all-sky fields. This suggests that cloud changes contribute substantially to the trend toward a less-negative net feedback for climates warmer than the PI, consistent with the kernel analysis results in Fig. 3.

This alternative approach also allows us to separate the influence of cloud shortwave effects from cloud longwave effects. We find that using clear-sky fields for only the longwave component of $\Delta N$ causes behavior resembling the all-sky results, whereas using clear-sky fields for only the shortwave component of $\Delta N$ causes behavior resembling the clear-sky results (SI Fig. S8). This suggests that the increase in the net feedback in warm climates is caused primarily by the cloud shortwave feedback, which is consistent with the results of previous studies[29,36,41].

**Planck feedback**. The Planck feedback describes how warming the surface and atmospheric column above causes more outgoing longwave radiation to space due to the Stefan-Boltzmann law. This feedback remains relatively invariant across the range of simulated climates, although it becomes slightly more negative as the climate cools. Note that because we use a radiative kernel, we account only for changes in the Planck feedback due to the evolving pattern of surface temperature change, and we do not represent how the Planck feedback depends on global temperature. The Planck feedback radiative kernel is most negative in the warmest regions of the control climate (see SI Section S3). The meridional structure of the surface temperature evolution is shown in SI Fig. S7. Simulated surface temperature changes tends to be amplified in ice-covered regions (Fig. 4f), which is expected to occur primarily due to the albedo feedback and lapse-rate feedback. As the ice-covered regions expand equatorward, the amplification moves out of the polar region, which causes the Planck feedback to become slightly more negative (see SI Section S3 for details). Note that Fig. 4f indicates that polar amplification is not a ubiquitous feature of climate change within this wide range of climates.

**Water-vapor feedback**. The water-vapor feedback occurs because warmer air can hold more water vapor, which is a greenhouse gas. This feedback tends to be more positive in warmer climates, for reasons that can be explained using idealized one-dimensional radiative-convective equilibrium models[31]. Consistent with this, we find that the strength of the water-vapor feedback varies approximately monotonically with the underlying climate, becoming more positive with warming, although it becomes fairly constant in climates warmer than the PI.

## Caveats

The results in Fig. 2 rely on direct model output in addition to the estimated $CO_2$ radiative forcing ($F_{GHG}$), which is computed using previously published line-by-line radiative transfer calculations[9]. These instantaneous radiative forcing (IRF) calculations do not account for stratospheric temperature adjustment, although they give similar results for our purposes to other previously published line-by-line radiative model results that do include stratospheric temperature adjustment[42] (SI Fig. S1). Neither calculation allows for the rapid adjustments to the tropospheric temperature profile in response to $CO_2$ forcing that are needed to estimate the effective radiative forcing (ERF)[43].

We assess the error associated with this approach by comparing with two separate estimates of the ERF associated with $CO_2$ quadrupling from the PI level in CESM2, noting that the error may be larger for climates farther from the PI. First, we use a preexisting CESM2 run[44] that has the sea-surface temperature (SST) field fixed at PI values and $CO_2$ increased by 4 × in order to estimate the ERF based on the change in TOA net radiation fields. Second, we use the regression method of Gregory[45] to estimate the ERF based on the first 20 years of a preexisting CESM2 simulation in which $CO_2$ was instantaneously quadrupled from its PI value[46]. In the latter analysis, the ERF is obtained by extrapolating the relationship between global TOA net energy flux and surface temperature to zero surface temperature anomaly. The results are 8.90 W/m² for the fixed-SST ERF estimate and 8.77 W/m² for the regression method ERF estimate, compared with 8.56 W/m² in the line-by-line radiative transfer code IRF estimate that we adopt in this analysis. The close agreement between the IRF estimate from the radiative transfer code and the ERF estimates from CESM2 may be coincidental given that CESM2, like most ESMs, shows substantial forcing adjustments from rapid changes in atmospheric temperature and cloud cover in response to $CO_2$ changes (e.g., ref. 47). However, this agreement gives confidence in the use of the IRF estimate (Fig. 2a) as an approximation to the ERF in CESM2 for our calculations.

Another consideration is whether radiative forcing should change with the underlying climate itself. Here we have adopted the standard definition of radiative forcing that assumes that $CO_2$ changes occur

within a constant climate (i.e., fixed surface temperature), and hence that all radiatively-important atmospheric and surface field changes beyond rapid adjustments are part of the radiative feedback on surface temperature changes. However, another defensible choice for the differential feedback would be to define radiative forcing relative to the continuously evolving climate, in which case the $CO_2$ forcing would change depending on factors including changes in atmospheric water vapor, cloud cover, and the difference in temperature between the surface and the stratosphere (e.g., refs. [48,49]). Calculating the radiative forcing under this alternative definition, which would require additional simulations, would modify the value of the differential feedback. Note that while this ambiguity in forcing definition is inherent to the differential feedback, the effective feedback only uses the standard radiative forcing definition adopted here because it is defined in terms of anomalies relative to the PI climate[43,48].

As noted above, the radiative kernel analysis does not allow the radiative response to perturbations in climate fields to evolve with the underlying climate because it effectively linearizes the simulated response about a climate near the PI. Furthermore, since the radiative kernels are set to zero above a fixed tropopause, radiative responses may not be calculated accurately in climates with a tropopause that is substantially higher than in the PI (e.g., ref. [31]).

To assess the accuracy of the kernel analysis, we re-ran the kernel analysis using clear-sky versions of the radiative kernels, which are included with the kernel fields[37]. The residual between the sum of the clear-sky feedback parameters and the clear-sky TOA net energy flux reported by the model is indicated as a red dashed line in the bottom row of Fig. 3. This provides an estimate of the uncertainty in the analysis. Although not negligible, the values are relatively small. Note that cancelation between feedbacks may play a role in these relatively small residuals (cf. ref. [50]), especially for climates far from the PI.

Furthermore, for the lapse-rate and albedo feedbacks, which dominate net feedback changes in colder climates, we found that using a horizontally-averaged kernel produced similar results. That is, horizontal variations in the kernel between the warm tropics and cold poles have minimal influence on how feedbacks change across climate states; instead, feedback changes primarily track changes in the global ice extent and the globally-averaged vertical structure of the atmosphere. This insensitivity to capturing differences in the radiative kernels across the range of spatial variations in the control climate (from the tropics to the poles) suggests that changes in the radiative efficiency of the atmosphere across climate states may be of secondary importance, supporting the accuracy of this analysis which uses a kernel that does not vary with climate.

This analysis uses an approximately equilibrated PI climate, whereas the simulated climates that are increasingly warmer or colder than the PI are expected to be increasingly far from equilibrium. Hence it may be seen as a source of concern that the net climate feedback is found to be most negative near the PI and increasingly less negative in climates increasingly warmer or colder than the PI. However, a number of factors suggest that the level of equilibration is not substantially influencing the values of the climate feedback that we calculate. First, we identify simple and fairly basic physical processes that drive the increase in sensitivity with cooling (related to the lapse rate and albedo feedbacks), suggesting that this is likely to be a robust climate response, and the increase in sensitivity with warming has been previously identified as a robust feature of many climate models (e.g., Fig. 7.11 of ref. [2]). Second, previous studies have found a loss of stability at the Snowball Earth bifurcation point, implying an increase in sensitivity as $\lambda_{net}^{diff}$ approaches zero under extreme cooling. Third, the minimum climate feedback is in a climate that is approximately 2K colder than the PI, rather than being at the equilibrated PI climate. Fourth, this approach does not depend on the level of equilibration, at least when applied to a simplified representation of the climate system (SI Fig. S3). Fifth, we find that the impact of deep ocean heat uptake efficacy would

not produce this shape (SI Fig. S4). And sixth, the clear-sky residual is relatively small (Fig. 3), showing that the alternative approach of using kernels, rather than the TOA balance used to generate the results in Fig. 2d, e, gives a similar result.

Moreover, we compared our results with a previous analysis[51] of a CESM2 instantaneous $CO_2$ quadrupling simulation[46]. Hahn et al.[51] used the same radiative kernels as the present study[37], and their results include values for the feedback parameters around simulation year 100 (averaged over their simulation years 85-115), at which point the global surface temperature is 6.6 °C above the initial PI value. We indicate their feedback parameter values at this level of warming as blue circles in Fig. 3. The agreement with our analysis (blue lines in Fig. 3) adds some confidence to our interpretation that the relationships we find between feedback values and global temperatures do not depend strongly on the degree of equilibration. We similarly included a blue circle indicating their value for $\lambda_{net}^{eff}$ in Fig. 2e, which agrees with our results (blue line in Fig. 2e).

Finally, the experimental design used here does not allow for slow feedbacks associated with factors including changes in ice sheets, the carbon cycle, and the deep ocean, which could modify the stability of the climate given sufficient time to adjust. These results should thus be interpreted as a measure of how the traditional fast feedbacks (i.e., Planck, water vapor, lapse rate, surface albedo, and clouds) depend on the underlying climate state, and they are relevant to studies that treat ice sheets and other slow feedbacks as external forcings (e.g., the LGM analysis of ref. [1]). If ice sheets were allowed to change, it is expected that their distinct spatial structure of ERF would produce different relationships between climate feedbacks and global temperature changes than those under $CO_2$ forcing alone explored here (e.g., refs. [52,53]). It is similarly expected that the results may differ if the model were allowed to approximately equilibrate to each level of $CO_2$, rather than using the 1% per year ramping adopted in the present study.

## Summary and conclusions

As constraints on the modern-day ECS based on past warm and cold climates gain in prominence (e.g., refs. [1,2,53]), it is becoming increasingly important to understand how and why climate feedbacks change over a wide range of climate states. In this study, we warmed and cooled a state-of-the-art climate model (NCAR CESM2) to simulate a continuum of climates ranging from a nearly ice-covered Snowball Earth to a nearly ice-free hothouse planet. We ramped $CO_2$ concentrations over a range of 11.5 doublings, which led to a 59 K range in simulated annual-mean global-mean transient surface temperature changes.

Previous studies have represented the dependence of climate feedbacks on the underlying global temperature by approximating that the net feedback scales linearly, which is equivalent to including a quadratic term in the global energy budget (e.g., ref. [1]). Our results suggest that this representation only approximately holds over a limited range of climates, spanning about 3K colder to 8K warmer than the PI climate. Importantly, LGM-like temperatures (4-6K colder than PI) fall outside of this range, suggesting that this representation is not accurate for assessing how LGM feedbacks relate to feedbacks in the modern-day or future climate, as has been done in previous analyses (e.g., ref. [1]). The "U" shape of the relationship we find between the net feedback and global temperature implies a stronger constraint lowering the upper bound of the ECS as inferred from LGM proxy reconstructions than reported by ref. [1], thereby implying a reduction in the uncertainty range for the ECS.

Since the relationship between the simulated net feedback and underlying climate is expected to depend on the choice of model, it would be useful to reproduce the present analysis using other ESMs. It is noteworthy that the 279-year and 514-year $CO_2$ ramping simulations generated for this analysis could be fairly straightforwardly repeated

with a different ESM. This would be particularly valuable because paleoclimate constraints on the ECS all rely on mapping feedbacks between different climate states. Recent studies using CESM2 identified an apparent cold bias in the simulation of the LGM climate[54] and warm bias in the simulation of the early Eocene[55], and a new version of the model was developed with cloud feedbacks tuned to be less positive ("CESM2-PaleoCalibr"[56]), which reduced the LGM bias and also resulted in a reduced modern-day ECS. Comparing the present analysis with a similar analysis that used CESM2-PaleoCalibr rather than CESM2 would further identify to what extent the tuning caused the dependence of the net feedback on the underlying climate to be shifted or restructured, which may shed further light on the way feedbacks in past climate states serve as analogs for feedbacks in the modern climate. That is, future work could determine whether identified biases in simulations of past warm climates using ESMs become reduced by changes in the value of the net feedback applying to all climates states (a vertical shift of the "U" shape in Fig. 2d, e) or by changes in the net feedback dependence on the underlying climate state (a change in the horizontal width of the "U" shape in Fig. 2d, e).

The results presented here are an initial step toward mapping feedback changes over a wide range of climates. They place past and future climate changes in a broader context, with implications for our understanding of what physical mechanisms cause the sensitivity of each radiative feedback to the underlying climate state.

## Methods
Here we describe details regarding the CESM2 simulations, how the $CO_2$ forcing was estimated, and how the feedback parameters were calculated.

### Simulation details
We use NCAR CESM2 in its standard workhorse configuration. The atmospheric component is CAM6, and the ocean component is POP2. The atmosphere and ocean both have nominal horizontal resolutions of 1°, and there are 32 vertical levels in the atmosphere and 60 vertical levels in the ocean.

The Warming and Cooling simulations are both branched from the end of year 500 of a previously run pre-industrial (PI) control simulation[7] with the forcing fixed at 1850 levels. The atmospheric $CO_2$ concentration is increased or decreased at a rate of 1% per year from the start of each simulation. For the first 150 years of the Warming run, we use the pre-existing CESM2 "1pctCO2" simulation that is part of the CMIP6 archive[8], which we extend to simulate further warming by branching to a cloned case. The Cooling run is identical to the Warming run except that the $CO_2$ change has the opposite sign.

Warming run details: This run initially failed during year 151 with the error "bounding bracket for pH solution not found" from `co2calc.F90`. Adjusting the POP time step from the default value `dt_count=48` to `dt_count=60` during years 151-152 caused this error to no longer occur. After year 279, there was an error in `lnd_import_export.F90` that the coupler was receiving an output of NaN from the land model. We were not able to resolve this error by reducing the CAM time step and ended the run after year 279.

Cooling run details: After year 278, this run failed with the error "bounding bracket for pH solution not found" from `co2calc.F90`, which was not resolved by increasing `dt_count`. We then commented out the line in the model code that called this error, which may lead to unreliable simulated pH. After year 332, when the $CO_2$ concentration reached approximately 10 ppm, the land component of the model failed with the error "CO2 is outside of an expected range" in `lnd_import_export.F90`, and we commented out the line in the model code that called this error. At the end of the 514-year run, there was an error with the iron flux being out of range in `marbl_diagnostics_mod.F90`, which we were not able to resolve

by simply commenting out the line in the model code that called this error.

Quantities analyzed: For $CO_2$, we use the atmospheric field `co2vmr`, which is the $CO_2$ volume mixing ratio. For surface temperature, we use the atmospheric field `TS`, which is the radiative surface temperature. For the measure of inversion strength in Fig. 4a, we compute the potential temperature from the atmospheric temperature `T` at vertical level 23, which is at approximately 700 hPa on the model hybrid vertical coordinate. For ice cover, we take the maximum of the fields `FSNO` and `PCT_GLACIER/100`, multiply this value by `landfrac`, and then add `ICEFRAC`. Here `FSNO` is the fraction of ground covered by snow reported by the land model, `PCT_GLACIER` is the percent of ground covered by glaciers which is included in the surface dataset input used by the land model, `landfrac` is the fraction of the grid box covered by land reported by the land model, and `ICEFRAC` is the fraction of the grid box covered by sea ice reported by the atmospheric model. We compute the net energy flux $N$ as `FSNT` − `FLNT`, with `FSNT` and `FLNT` the top-of-model net longwave and solar fluxes reported by the atmospheric model.

### $CO_2$ forcing
Byrne and Goldblatt[9] used a line-by-line radiative transfer code to calculate forcing from $CO_2$ (as well as other greenhouse gases). The publication includes a supplemental data text file ("text03.txt") that has radiative forcing associated with $CO_2$ concentrations varying from 1 ppm to 100,000 ppm. Although this is a considerably wider range of $CO_2$ concentrations than mentioned in their actual paper, the supplemental data values are valid output from their radiative model (Brendan Byrne, personal communication, January 2021).

The $CO_2$ in our simulations ranges from 1.6 ppm to 3422 ppm. We calculate the associated radiative forcing $F_{GHG}$ using a cubic interpolation of the relationship between the radiative forcing associated with the annual-mean global-mean profile ("GAM" in "text03.txt") and the logarithm of the $CO_2$ concentration ("CO2" in "text03.txt"), which is shown in SI Fig. S1.

### Calculation of effective and differential feedback parameters
For the effective feedback parameters, we smooth each radiative response time series ($F_{net}$ or $F_i$) using a least-squares fit to a 12th-order polynomial in $(T − T_0)$ that is constrained to go through $(T_0, F_0)$, where $T_0$ and $F_0$ are the surface temperature and radiative response ($F_{net}$ or $F_i$) averaged over years 480-499 of the PI simulation. This allows the ratio in Eq. (2) to be smooth even in the limit $T \to T_0$. This smoothing of $F_{net}$, and the resulting values of $\lambda_{net}^{\mathrm{eff}}$ and EffCS, are plotted in SI Fig. S2 next to the raw unsmoothed annual-mean simulation output.

For the differential feedback parameters, we regress the radiative response ($F_{net}$ or $F_i$) on the surface temperature $T$. We use a total-least-squares (TLS) regression, rather than a standard ordinary-least-squares (OLS) regression, because the radiative response ($F_{net}$ or $F_i$) and temperature ($T$) both play the role of dependent variables. A TLS regression accounts for errors in both variables, whereas an OLS regression accounts for errors in one variable and treats the other as an independent variable. The TLS regression depends on the choice of units, and we normalize each variable by the standard deviation of the residuals of the time trend, following Winton[57]. We compute the TLS regression in a running window of variable duration that spans temperatures in the range $\pm 3K$.

## Data availability
Model output from the Warming and Cooling simulations is available at https://eisenman-group.github.io. The kernels used in this analysis were downloaded from https://github.com/apendergrass/cam5-kernels. Source data for the line plots in Figs. 1–4 are provided with this paper.

## Code availability

Code to compute the differential and effective net feedback parameters (Fig. 2d,e) from the simulation output, which can similarly be used with the kernels to compute the individual feedback parameters (Fig. 3), is available at https://eisenman-group.github.io.

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

## Acknowledgements

Without implying their endorsement, we thank Vince Cooper, Lily Hahn, Jack Bauchop, Ivan Mitevski, Jonah Bloch-Johnson, Brian Rose, Nadir Jeevanjee, Nick Lutsko, Matt Long, David Neelin, Angie Pendergrass, and Brendan Byrne for helpful discussions at various points during the course of this work. This work was supported by US National Science Foundation grants OCE-2048590 (IE), AGS-1752796 (KCA), and OCE-2002276 (KCA), and National Oceanic and Atmospheric Administration MAPP Program award NA20OAR4310391 (KCA).

## Author contributions

K.C.A. and I.E. conceived the study. I.E. carried out the simulations. I.E. and K.C.A. designed and carried out the analysis, interpreted the results, and wrote the manuscript.

## Competing interests

The authors declare no competing interests.

## Additional information

**Peer review information** : *Nature Communications* thanks the anonymous reviewers for their contribution to the peer review of this work. A peer review file is available.

