## [Peer Review File · Nature Communications]

Reviewers' Comments:

Reviewer #1:

Remarks to the Author:

This paper performs idealized simulations using an Earth System Model (CESM2) with large increases or decreases of the atmospheric CO₂ to map the climate sensitivity across a wide range of warm and cold climates. The authors' simulations show a U-shaped climate sensitivity with a stability optimum near the PI, meaning climate sensitivity increases for both sufficiently warmer and colder climates. The authors use radiative kernels to understand the behavior and find that the albedo and lapse rate feedbacks are responsible for the increased sensitivity in cold climates and that the cloud and lapse-rate feedbacks explain the increased sensitivity in warm climates. The authors suggest that assumptions of a constant or a linear relationship between climate sensitivity and the global temperature that have been used in previous studies (such as Sherwood et al. 2020), could lead to biased estimation of climate sensitivity.

This is a very interesting study using a simple but novel approach to show an important aspect of the climate sensitivity: its state dependence. A better understanding of the state dependence of climate sensitivity has huge implication for the understanding of past warm and cold climates and for the use of paleoclimate to constrain climate sensitivity. This study presents a welcoming step toward mapping climate sensitivity and feedbacks over a wide range of climate.

Despite the known problems of CESM2 (see my comments below), I think the authors result (a U-shaped climate sensitivity) is robust, mostly because the key processes for the increase of sensitivity with cooling rely on simple and basic physics (ice albedo feedback) and thus should not be qualitatively impacted by CESM2's problems in parameterizations (e.g., shown in Zhu et al. 2022). In addition, an increase of climate sensitivity with warming seems to be a robust feature of many climate models (e.g., see IPCC AR6 Figure 7.11). Given this and the fact the authors have provide discussion on the caveats of their work, I support the publication of the paper in Nature Communications. Below are a few comments for the authors to consider.

1. The caveats on CESM2's high climate sensitivity should be expanded. It has been shown that CESM2 substantially overestimates the global warming during the early Eocene and the global cooling during the LGM (Zhu et al., 2020;2021). How will CESM2's unrealistically high ECS impact the results in the study? Will the U-shape be broader if a low-ECS model (e.g., CCSM4 or CESM1(CAM5)) is used? Will the height of the edge of the "U" be impacted?
2. The authors could consider expanding the discussion of the state dependence of feedbacks. For example, are the increase of cloud and lapse rate feedbacks with warming robust? What are the physical processes driving the increases? The authors seem to suggest that mechanisms are beyond the scope of the paper, but it would be nice to briefly provide some mechanistic understanding. If these hypotheses are rooted in simple physical processes, the authors findings are more likely to be robust (less likely to be impacted by the uncertain physical parameterizations). Also, this discussion could be done in the context of many previous studies on the state dependence of climate forcing and feedbacks, such as Meraner et al. (2013), Caballero and Huber (2013), and Zhu et al. (2019).
3. Please consider providing some quantitative estimation on how much the linear relationship between climate sensitivity and the global temperature could impact the estimation of ECS (e.g., in Sherwood et al., 2020). I understand that there are a lot of uncertainties involved, in particular the single-model approach of the present study, but I think some estimation (or speculation) on the magnitude of the impact would be nice for readers to evaluate the importance of the authors finding.
4. Line 333: "Cloud feedbacks are more destabilizing in warm climates" has been found in multiple versions of CESM, including the CCSM3 in Caballero and Huber (2013), CESM1 in Zhu et al.

(2019), and the CCSM4, CESM1, and CESM2 in Zhu and Poulsen (2020).

References:

Caballero, R., & Huber, M. (2013). State-dependent climate sensitivity in past warm climates and its implications for future climate projections. *Proceedings of the National Academy of Sciences*, 110(35), 14162–14167. <https://doi.org/10.1073/pnas.1303365110>

Meraner, K., Mauritsen, T., & Voigt, A. (2013). Robust increase in equilibrium climate sensitivity under global warming. *Geophysical Research Letters*, 40(22), 5944–5948. <https://doi.org/10.1002/2013GL058118>

Zhu, J., Poulsen, C. J., & Tierney, J. E. (2019). Simulation of Eocene extreme warmth and high climate sensitivity through cloud feedbacks. *Science Advances*, 5(9), eaax1874. <https://doi.org/10.1126/sciadv.aax1874>

Zhu, J., & Poulsen, C. J. (2020). On the Increase of Climate Sensitivity and Cloud Feedback With Warming in the Community Atmosphere Models. *Geophysical Research Letters*, 47(18), e2020GL089143. <https://doi.org/10.1029/2020GL089143>

Zhu, J., Poulsen, C. J., & Otto-Bliesner, B. L. (2020). High climate sensitivity in CMIP6 model not supported by paleoclimate. *Nature Climate Change*, 10(5), 378–379. <https://doi.org/10.1038/s41558-020-0764-6>

Zhu, J., Otto-Bliesner, B. L., Brady, E. C., Poulsen, C. J., Tierney, J. E., Lofverstrom, M., & DiNezio, P. (2021). Assessment of equilibrium climate sensitivity of the Community Earth System Model version 2 through simulation of the Last Glacial Maximum. *Geophysical Research Letters*, n/a(n/a), e2020GL091220. <https://doi.org/10.1029/2020GL091220>

Reviewer #2:

Remarks to the Author:

General comments

The authors estimate the temperature-dependence of the climate feedback parameter—and related climate sensitivity. They do so by forcing a state-of-the-art GCM with a wide range of CO₂ concentrations to span surface temperatures from -25 to almost 30 degrees. A regression of energy imbalance against surface temperatures is used to quantify the feedback at different base climates. In addition, the authors try to disentangle the proportion of different feedback components using a radiative kernel technique.

The main finding is that the climate sensitivity reaches a local minimum at surface temperatures close to present-day. This finding is in general agreement with idealised modelling studies [Seeley and Jeevanjee (2020), Kluft et al. (2021)]. This study adds two novel aspects: i) the GCM seems to reach the local minimum at lower temperatures than the idealised models, and ii) the authors point out that many studies dealing with the temperature-dependency of the feedback parameters have chosen a linear-approach that is not in agreement with this finding (and the idealised studies).

In the second half of the manuscript, the authors try to associate the temperature-dependence of the feedback parameter with individual processes (feedbacks) by using radiative kernels. Although I like the general idea, I find this part of the manuscript not convincing. Radiative kernels are strongly dependent on the base climate they have been calculated in. This dependence can be linearised to a certain degree, but to me the results presented in Fig. 3 indicate that the kernels are not suitable for this large range of surface temperatures. My main concerns are:

1. The almost constancy of the Planck feedback is in stark contrast to the expected T^3 dependence. This difference is likely caused by the fact that it is not represented in the kernels (L263-264).
2. The very muted increase of the water-vapour feedback at high temperatures (L313-314)). Many line-by-line modelling studies [Koll and Cronin (2018), Seeley and Jeevanjee (2020), Kluft et al. (2021)] as well as more analytical approaches [Simpson (1928), Nakajima et al., 1992), Ingram (2010)] point towards a rapidly declining water-vapour feedback as soon as the atmospheric emission window shuts due to increased continuum absorption at high surface temperatures.
3. The imbalance of water-vapor and lapse-rate feedbacks. Usually, one would assume that these two feedbacks cancel each other quite well (as seen for surface temperatures between 5 and 20 degrees). This cancellation vanishes at colder temperatures, which the authors interpret as a more stabilising lapse-rate feedback. An alternative explanation would be that the linear extrapolation of the radiative kernels is not sufficient anymore: an increase/decrease of upper-tropospheric temperatures is thought to be accompanied by a moistening/drying of the same atmosphere layers; an effect that is likely not captured in the kernels.

The authors defend the usage of the kernels in two different ways. First, they compare their results to another study which used the same radiative kernels. This gives confidence in the proper application of the kernels, but not in the kernels themselves. Second, they compute the residual between the net clear-sky TOA simulated by the GCM to results obtained using the clear-sky radiative kernels. This ought to simultaneously prove the accuracy of the cloud feedback (L340-384) and the usage of the kernels in general (L399-403). What I don't understand is, why this residual is almost at its maximum for the PI climate and apparently becomes negligible at temperatures far below the reference climate.

Despite my concerns about the methodology used for decomposing the feedback parameter, the study's findings regarding the local minimum in ECS challenge some widespread assumptions and offer a valuable extension to the work on more idealised models. However, I think it is necessary to address these concerns and refine the methodology to further strengthen the impact of this research.

Minor comments

L82 28.7C is not 14.8C degrees warmer than the 15.0C in PI.

L465-466 In my opinion, the relationship between feedback and temperature should depend on physical processes and therefore be similar across models? Could this be rephrased as how varying feedbacks across GCMs could be linked to their different base climates?

L683-684 Is this sufficient to keep the model close to an equilibrated state? In my understanding this would be necessary to estimate the feedbacks from the TOA imbalance.

L699 It would be nice to know what the modifications were. Otherwise the model differs from NCAR CESM2 in a way that is not transparent.

L356-392 and L715-726 I agree that the definition of the instantaneous forcing allows changes in the water-vapour concentration to be considered as feedbacks rather the forcing. However, the state-dependence of the CO₂ forcing is rather large at these CO₂ and T ranges. Is it possible to estimate how large the additional "feedback" is and in which of the kernel/feedback it would be visible?

Reviewer #3:

Remarks to the Author:

Review of "The radiative feedback continuum from Snowball Earth to an ice-free hothouse"

Authors: I. Eisenman and K. C. Armour

In this study the authors explore how climate sensitivity varies as the earth warms or cools. Past studies indicate that climate sensitivity increases as the earth warms and decreases for a slight cooling. It is also known that were the earth cool enough, climate would become unstable, and snow and ice would cover the entire earth's surface. Finally, the concept of a runaway greenhouse warming is not new and indicates that the climate of a much warmer earth would also be unstable with the Earth losing all of its water (as did Venus). So, it isn't particularly surprising that the present climate state is more stable than both warmer and colder climate states. The current study indicates that the present climate is more stable than all other states except those a few degrees cooler. Quantification with the CESM2 model of climate stability as a function of global mean temperature is a new result.

The analysis is sound and the exposition is clear. Thank you! The results reported are novel in that they indicate the limits to the approximation that climate sensitivity depends linearly on temperature, which the authors show is valid only in the range 3K cooler to 10K warmer than present day. This makes it more difficult to directly infer from paleoclimates (with temperatures cooler by more than 4K) the climate sensitivity of today's climate. To the authors' credit, several caveats are discussed, which help the reader interpret which aspects of the results may not be robust.

I have two suggestions for improving the manuscript. First, I would eliminate panel f of figure 2, which is a plot of "effective climate sensitivity" as a function of temperature. The curve shown is proportional to the negative inverse of the curve in panel e, and thus it is unnecessary since it provides no additional information. More importantly, the interpretation of "effective climate sensitivity" as defined for this figure is problematic. "Effective climate sensitivity", which is the equilibrium temperature change expected were atmospheric CO₂ concentration doubled, has been seen by some as a more "accessible" measure of climate sensitivity than the more fundamental and useful "feedback parameter", which also characterizes sensitivity. But in the context of this study, it makes little sense to plot the effective climate sensitivity as a function of temperature. Consider, for example, a climate some 35K cooler than the present (i.e., at ~-20K). At that temperature, the authors show the earth is plunging toward a snowball condition, and if CO₂ were doubled, even instantaneously, this would not prevent earth ending up in an even cooler state. Yet, panel f indicates that the earth should after doubling of CO₂ end up about 10K warmer than its initial temperature (i.e., reach an equilibrium temperature of about -10K). We know that at -10K the earth will still eventually become a "snowball", so the contention that panel f is useful in indicating "effective climate sensitivity" is unsupported. Besides deleting panel f, the discussion of it and of "effective climate sensitivity" would also, of course, need to be eliminated.

The second suggestion involves the claim in the concluding section that the results are important because they indicate that reliance on paleoclimate estimates of climate sensitivity (for temperatures 2K cooler than pre-industrial) cannot constrain current model estimates of climate sensitivity. More discussion and qualification of this conclusion are needed. In particular, the statement that "net feedback (and climate stability) change relatively rapidly for temperatures more than 2K colder than the PI" isn't supported by figure 2, which by eye seems to indicate the climate sensitivity remains within about 10% of its present value even for climates as much as 10K cooler than present. Since the range of climate sensitivities estimated by models is much larger than this, I think the paleoclimate data may still be of substantial value. The authors should therefore be more circumspect in their discussion of the implications (and practical significance) of their results.

Besides the above two suggestions, I have the following comments and suggestions:

Lines 76-86: In the discussion of "ice cover" and its changes, you should mention here that glacial ice amounts were prescribed and held fixed.

Line 118: You need to indicate how "effective feedback" is estimated for PI conditions (and plotted on fig. 2e) since it is undefined when ΔT is 0. I guess you must interpolate, but you should mention this.

Eq. 4: a plus sign is missing between the first two terms on the RHS.

Line 219: replace "atmospheric temperature" with "atmospheric lapse rate".

Line 237: indicates that "cloud and lapse-rate feedbacks" are primarily responsible for the decreased stability as climate warms. It looks to me like the contribution from the residual might be as significant as lapse-rate/water vapor feedback, and this should probably be noted.

Line 321: I don't think subsections are labeled in the SI, so replace "Sec. 5.2" with "Sec. 5".

Lines 404-416: Yes, the Hahn et al. (2021) results seem consistent with the present study, but I'm not sure this tells us much about whether the kernels can be held fixed, independent of climate state. And the fact that the clear-sky residuals are not too large also isn't informative in this regard. How do you know that the kernels don't strongly depend on climate state (especially as climate becomes substantially different from the present one)?

Line 420: replace "Cooling" with "cooling".

Line 441: replace "was" with "were".

Line 451: replace "lead" with "led" and replace "surface temperature" with "transient surface temperature changes".

Responses to Reviewer #1

We thank the reviewer for these helpful comments. We have addressed them as follows in the revised manuscript (reviewer's comments in italics):

This paper performs idealized simulations using an Earth System Model (CESM2) with large increases or decreases of the atmospheric CO₂ to map the climate sensitivity across a wide range of warm and cold climates. The authors' simulations show a U-shaped climate sensitivity with a stability optimum near the PI, meaning climate sensitivity increases for both sufficiently warmer and colder climates. The authors use radiative kernels to understand the behavior and find that the albedo and lapse rate feedbacks are responsible for the increased sensitivity in cold climates and that the cloud and lapse-rate feedbacks explain the increased sensitivity in warm climates. The authors suggest that assumptions of a constant or a linear relationship between climate sensitivity and the global temperature that have been used in previous studies (such as Sherwood et al. 2020), could lead to biased estimation of climate sensitivity.

This is a very interesting study using a simple but novel approach to show an important aspect of the climate sensitivity: its state dependence. A better understanding of the state dependence of climate sensitivity has huge implication for the understanding of past warm and cold climates and for the use of paleoclimate to constrain climate sensitivity. This study presents a welcoming step toward mapping climate sensitivity and feedbacks over a wide range of climate.

Despite the known problems of CESM2 (see my comments below), I think the authors result (a U-shaped climate sensitivity) is robust, mostly because the key processes for the increase of sensitivity with cooling rely on simple and basic physics (ice albedo feedback) and thus should not be qualitatively impacted by CESM2's problems in parameterizations (e.g., shown in Zhu et al. 2022). In addition, an increase of climate sensitivity with warming seems to be a robust feature of many climate models (e.g., see IPCC AR6 Figure 7.11). Given this and the fact the authors have provide discussion on the caveats of their work, I support the publication of the paper in Nature Communications. Below are a few comments for the authors to consider.

Reply: We appreciate this thoughtful and generous summary of our manuscript.

1. The caveats on CESM2's high climate sensitivity should be expanded. It has been shown that CESM2 substantially overestimates the global warming during the early Eocene and the global cooling during the LGM (Zhu et al., 2020;2021). How will CESM2's unrealistically high ECS impact the results in the study? Will the U-shape be broader if a low-ECS model (e.g., CCSM4 or CESM1(CAM5)) is used? Will the heigh of the edge of the "U" be impacted?

Reply: We appreciate this comment and have expanded our discussion of this topic in the revised manuscript. Although we do not think we can confidently assess how the U-shape would differ in another model without carrying out additional simulations, we thank the reviewer for this framing of the question in terms of the width and height of the U-shape, which we have incorporated into the revised manuscript. The expanded discussion includes the following (lines 555-572):

Since the relationship between the simulated net feedback and underlying climate is expected to depend on the choice of model, it would be useful to reproduce the present

analysis using other ESMs. This would be particularly valuable because paleoclimate constraints on the ECS all rely on mapping feedbacks between different climate states. Recent studies using CESM2 identified an apparent cold bias in the simulation of the LGM climate (Zhu et al., 2021) and warm bias in the simulation of the early Eocene (Zhu et al., 2022), and a new version of the model was developed with cloud feedbacks tuned to be less positive (“CESM2-PaleoCalibr”, Zhu et al., 2022), which reduced the LGM bias and also resulted in a reduced modern-day ECS. Comparing the present analysis with a similar analysis that used CESM2-PaleoCalibr rather than CESM2 would further identify to what extent the tuning caused the dependence of the net feedback on the underlying climate to be shifted or restructured, which may shed further light on the way feedbacks in past climate states serve as analogs for feedbacks in the modern climate. That is, future work could determine whether identified biases in simulations of past warm climates using ESMs become reduced by changes in the value of the net feedback applying to all climates states (a vertical shift of the “U” shape in Figs. 2d,e) or by changes in the net feedback dependence on the underlying climate state (a change in the horizontal width of the “U” shape in Figs. 2d,e).

2. The authors could consider expanding the discussion of the state dependence of feedbacks. For example, are the increase of cloud and lapse rate feedbacks with warming robust? What are the physical processes driving the increases? The authors seem to suggest that mechanisms are beyond the scope of the paper, but it would be nice to briefly provide some mechanistic understanding. If these hypotheses are rooted in simple physical processes, the authors findings are more likely to be robust (less likely to be impacted by the uncertain physical parameterizations). Also, this discussion could be done in the context of many previous studies on the state dependence of climate forcing and feedbacks, such as Meraner et al. (2013), Caballero and Huber (2013), and Zhu et al. (2019).

Reply: In the revised manuscript, we have focused the discussion more clearly on the increase in sensitivity in cold climates, in order to keep the emphasis on the most novel results. We have added text to clarify this, including (lines 318-320):

The decrease in stability with cooling in cold climates is the main novel result of the present study, since previous work has discussed the decrease in stability with warming. Hence we begin by interpreting the lapse-rate and albedo feedbacks.

Following the reviewer’s suggestion, we now point readers to other papers in our discussion of the cloud feedback (lines 379-382):

Previous work using CESM2 and earlier versions of this model similarly found that cloud feedbacks are more destabilizing in warmer climates (Caballero & Huber, 2013; Zhu et al., 2019; Zhu & Poulsen, 2020).

Furthermore, in the revised manuscript we note that our finding that the cloud feedback changes are primarily due to the cloud shortwave feedback is “consistent with the results of previous studies (Caballero & Huber, 2013; Zhu et al., 2019; Zhu & Poulsen, 2020)” (lines 403-405). We also discuss changes in the lapse-rate feedback in climates warmer than the PI, noting (lines 340-342):

An analysis of a previous version of this model lead to fairly similar changes in the spatial pattern of the lapse-rate feedback parameter under varied levels of forced warming (Merlis et al., 2022).

We hope these revisions are sufficient to address the reviewer’s comment.

3. *Please consider providing some quantitative estimation on how much the linear relationship between climate sensitivity and the global temperature could impact the estimation of ECS (e.g., in Sherwood et al., 2020). I understand that there are a lot of uncertainties involved, in particular the single-model approach of the present study, but I think some estimation (or speculation) on the magnitude of the impact would be nice for readers to evaluate the importance of the authors finding.*

Reply: We have added a brief discussion regarding the implied change in the Sherwood et al. (2020) result (lines 236-249):

These results show that the value of α adopted by Sherwood et al. (2020) for the change in λ_{net} at the LGM is much larger than in the CESM2 results, because the “U” shape in Figs. 2d,e causes the feedback at 5K of cooling to be similar to the feedback at the PI. If we were to repeat the Sherwood et al. (2020) analysis using the value of α that we find here for the difference between the LGM and PI feedbacks, our lower value of α would imply a lower modern-day climate sensitivity than Sherwood et al. (2020) found, which amounts to a stronger constraint on the upper bound of the EffCS than they report. This is because the CESM2 results suggest that the LGM may be a more direct analogue to current warming than previously assumed, since the feedbacks are relatively similar. In other words, Sherwood et al. (2020) took λ_{net} to be more negative at the LGM than the modern value, whereas we find that the feedbacks are similar. So a given paleo estimate of the LGM value of λ_{net} implies a similar modern feedback value according to our results, whereas the analysis of Sherwood et al. (2020) would take it to imply a less negative modern feedback and hence a more sensitive modern climate.

4. *Line 333: “Cloud feedbacks are more destabilizing in warm climates” has been found in multiple versions of CESM, including the CCSM3 in Caballero and Huber (2013), CESM1 in Zhu et al. (2019), and the CCSM4, CESM1, and CESM2 in Zhu and Poulsen (2020).*

Reply: We appreciate the reviewer bringing this up. As also noted above, we have added the following sentence (lines 379-382):

Previous work using CESM2 and earlier versions of this model similarly found that cloud feedbacks are more destabilizing in warmer climates (Caballero & Huber, 2013; Zhu et al., 2019; Zhu & Poulsen, 2020).

Responses to Reviewer #2

We thank the reviewer for these helpful comments. We have addressed them as follows in the revised manuscript (reviewer's comments in italics):

General comments

The authors estimate the temperature-dependence of the climate feedback parameter—and related climate sensitivity. They do so by forcing a state-of-the-art GCM with a wide range of CO₂ concentrations to span surface temperatures from -25 to almost 30 degrees. A regression of energy imbalance against surface temperatures is used to quantify the feedback at different base climates. In addition, the authors try to disentangle the proportion of different feedback components using a radiative kernel technique.

The main finding is that the climate sensitivity reaches a local minimum at surface temperatures close to present-day. This finding is in general agreement with idealised modelling studies [Seeley and Jeevanjee (2020), Kluft et al. (2021)]. This study adds two novel aspects: i) the GCM seems to reach the local minimum at lower temperatures than the idealised models, and ii) the authors point out that many studies dealing with the temperature-dependency of the feedback parameters have chosen a linear-approach that is not in agreement with this finding (and the idealised studies).

In the second half of the manuscript, the authors try to associate the temperature-dependence of the feedback parameter with individual processes (feedbacks) by using radiative kernels. Although I like the general idea, I find this part of the manuscript not convincing. Radiative kernels are strongly dependent on the base climate they have been calculated in. This dependence can be linearised to a certain degree, but to me the results presented in Fig. 3 indicate that the kernels are not suitable for this large range of surface temperatures.

Reply: We appreciate this summary of our manuscript.

See our comment below regarding Fig. 3, which had an error in the original submission.

My main concerns are:

1. The almost constancy of the Planck feedback is in stark contrast to the expected T^3 dependence. This difference is likely caused by the fact that it is not represented in the kernels (L263-264).

Reply: The reviewer is correct. We point this out on lines 409-412:

Note that because we use a radiative kernel, we account only for changes in the Planck feedback due to the evolving pattern of surface temperature change, and we do not represent how the Planck feedback depends on global temperature.

Similarly, we also explain (lines 477-479):

Note that cancelation between feedbacks may play a role in these relatively small residuals (cf. Koll & Cronin, 2018), especially for climates far from the PI.

2. The very muted increase of the water-vapour feedback at high temperatures (L313-314)). Many

line-by-line modelling studies [Koll and Cronin (2018), Seeley and Jeevanjee (2020), Kluft et al. (2021)] as well as more analytical approaches [Simpson (1928), Nakajima et al., 1992), Ingram (2010)] point towards a rapidly declining water-vapour feedback as soon as the atmospheric emission window shuts due to increased continuum absorption at high surface temperatures.

Reply: We appreciate the reviewer raising this point and have added a brief discussion of it, clarifying that previous studies have identified this effect in climates warmer than we simulate (lines 176-179):

Note that previous studies using idealized single-column radiative models have found that the net climate feedback becomes more negative with warming for climates warmer than approximately 40°C (Seeley & Jeevanjee, 2021; Kluft et al., 2021).

By contrast, we have focused the discussion in the revised manuscript more clearly on the increase in sensitivity in cold climates, in order to keep emphasis on the most novel results, explaining (lines 318-320):

The decrease in stability with cooling in cold climates is the main novel result of the present study, since previous work has discussed the decrease in stability with warming. Hence we begin by interpreting the lapse-rate and albedo feedbacks.

3. The imbalance of water-vapor and lapse-rate feedbacks. Usually, one would assume that these two feedbacks cancel each other quite well (as seen for surface temperatures between 5 and 20 degrees). This cancellation vanishes at colder temperatures, which the authors interpret as a more stabilising lapse-rate feedback. An alternative explanation would be that the linear extrapolation of the radiative kernels is not sufficient anymore: an increase/decrease of upper-tropospheric temperatures is thought to be accompanied by a moistening/drying of the same atmosphere layers; an effect that is likely not captured in the kernels.

Reply: The reviewer is correct that the water-vapor and lapse-rate feedbacks largely cancel for climates similar to the reference climate on which the kernel is based (red dashed lines in third row of Fig. 3), whereas there is considerably less cancellation for colder climates, which could be seen as a cause for concern. However, the relatively small clear-sky residual (red dashed lines in bottom row of Fig. 3) implies that if the kernel is missing a compensating effect between these two feedbacks, then it is doing this in a way that the total clear-sky feedback is still relatively well represented.

The authors defend the usage of the kernels in two different ways. First, they compare their results to another study which used the same radiative kernels. This gives confidence in the proper application of the kernels, but not in the kernels themselves. Second, they compute the residual between the net clear-sky TOA simulated by the GCM to results obtained using the clear-sky radiative kernels. This ought to simultaneously prove the accuracy of the cloud feedback (L340-384) and the usage of the kernels in general (L399-403). What I don't understand is, why this residual is almost at its maximum for the PI climate and apparently becomes negligible at temperatures far below the reference climate.

Reply: It turns out that I inadvertently overwrote a variable in the code to make the final version of

Fig. 3, which caused the wrong quantity to be plotted for the clear-sky residual in the bottom row of Fig. 3, and I somehow failed to notice that the final version of the figure had this error. I am extremely grateful to the reviewer for catching this! I have corrected it in the revised manuscript, which shows a considerably smaller clear-sky residual in Fig. 3.

Despite my concerns about the methodology used for decomposing the feedback parameter, the study's findings regarding the local minimum in ECS challenge some widespread assumptions and offer a valuable extension to the work on more idealised models. However, I think it is necessary to address these concerns and refine the methodology to further strengthen the impact of this research.

Reply: We appreciate this comment and hope that our revisions, as discussed above, have sufficiently addressed the reviewer's concerns.

Minor comments

L82 28.7C is not 14.8C degrees warmer than the 15.0C in PI.

Reply: We appreciate the reviewing catching this embarrassing typo! We have corrected it. It now reads that the PI temperature of “15°C” (line 79) “increases by 15K to 30°C” (line 82). (We rounded these values to the nearest integer for consistency with other reported temperatures in the revised manuscript.)

L465-466 In my opinion, the relationship between feedback and temperature should depend on physical processes and therefore be similar across models? Could this be rephrased as how varying feedbacks across GCMs could be linked to their different base climates?

Reply: The reviewer raises an interesting point (as we understand it): that if there are simple physical processes linking feedback parameter values to the mean state climate, as we're suggesting, then this could also have implications for understanding feedback differences across climate models that simulate different mean states. We have not looked into this question but see it as a potentially promising topic to investigate in a future study. At the same time, though, the level of inter-model differences in their mean states is relatively small compared with the range of climates that we are looking at. The mean states all essentially hover around the bottom of the “U” that we find, which gives us less confidence that the impact of mean state differences in the context of this study would explain much of the inter-model differences in feedback strengths.

L683-684 Is this sufficient to keep the model close to an equilibrated state? In my understanding this would be necessary to estimate the feedbacks from the TOA imbalance.

Reply: No, the 1%/yr ramping is not sufficiently slow, and the model is expected to become increasingly out of equilibrium as CO₂ is ramped from the PI level. To clarify this, in the revised manuscript we note that “the simulated climates that are increasingly warmer or colder than the PI are expected to be increasingly far from equilibrium” (lines 490-492). However, our analysis does not assume that the climate system is near equilibrium. This is a subtle point, and we have added a substantial new discussion to clarify this in the revised manuscript on lines 875-913 in the SI and Fig. S3, including (lines 904-905): “This helps to illustrate how the analysis used in this study does not depend on how equilibrated the climate system is with the evolving value of ΔF_{GHG} .” We have also explained in the main text (lines 504-506): “this approach does not depend on the level of equilibration, at least

when applied to a simplified representation of the climate system (SI Fig. S3)’.’

L699 It would be nice to know what the modifications were. Otherwise the model differs from NCAR CESM2 in a way that is not transparent.

Reply: The modification to the code was simply to comment out a line. We have clarified this in the revised manuscript as follows (lines 818-819): “we commented out the line in the model code that called this error, which may lead to unreliable simulated pH.”

L356-392 and L715-726 I agree that the definition of the instantaneous forcing allows changes in the water-vapour concentration to be considered as feedbacks rather the forcing. However, the state-dependence of the CO2 forcing is rather large at these CO2 and T ranges. Is it possible to estimate how large the additional “feedback” is and in which of the kernel/feedback it would be visible?

Reply: This is an interesting question that we think deserves more study, although we expect that we would need to use fixed-SST simulations to evaluate it, and we see it as outside the scope of the present study.

Responses to Reviewer #3

We thank the reviewer for these helpful comments. We have addressed them as follows in the revised manuscript (reviewer's comments in italics):

Review of “The radiative feedback continuum from Snowball Earth to an ice-free hothouse”

Authors: I. Eisenman and K. C. Armour

In this study the authors explore how climate sensitivity varies as the earth warms or cools. Past studies indicate that climate sensitivity increases as the earth warms and decreases for a slight cooling. It is also known that were the earth cool enough, climate would become unstable, and snow and ice would cover the entire earth's surface. Finally, the concept of a runaway greenhouse warming is not new and indicates that the climate of a much warmer earth would also be unstable with the Earth losing all of its water (as did Venus). So, it isn't particularly surprising that the present climate state is more stable than both warmer and colder climate states. The current study indicates that the present climate is more stable than all other states except those a few degrees cooler. Quantification with the CESM2 model of climate stability as a function of global mean temperature is a new result.

The analysis is sound and the exposition is clear. Thank you! The results reported are novel in that they indicate the limits to the approximation that climate sensitivity depends linearly on temperature, which the authors show is valid only in the range 3K cooler to 10K warmer than present day. This makes it more difficult to directly infer from paleoclimates (with temperatures cooler by more than 4K) the climate sensitivity of today's climate. To the authors' credit, several caveats are discussed, which help the reader interpret which aspects of the results may not be robust.

Reply: We appreciate this thoughtful and generous summary of the manuscript.

I have two suggestions for improving the manuscript. First, I would eliminate panel f of figure 2, which is a plot of “effective climate sensitivity” as a function of temperature. The curve shown is proportional to the negative inverse of the curve in panel e, and thus it is unnecessary since it provides no additional information. More importantly, the interpretation of “effective climate sensitivity” as defined for this figure is problematic. “Effective climate sensitivity”, which is the equilibrium temperature change expected were atmospheric CO₂ concentration doubled, has been seen by some as a more “accessible” measure of climate sensitivity than the more fundamental and useful “feedback parameter”, which also characterizes sensitivity. But in the context of this study, it makes little sense to plot the effective climate sensitivity as a function of temperature. Consider, for example, a climate some 35K cooler than the present (i.e., at -20K). At that temperature, the authors show the earth is plunging toward a snowball condition, and if CO₂ were doubled, even instantaneously, this would not prevent earth ending up in an even cooler state. Yet, panel f indicates that the earth should after doubling of CO₂ end up about 10K warmer than its initial temperature (i.e., reach an equilibrium temperature of about -10K). We know that at -10K the earth will still eventually become a “snowball”, so the contention that panel f is useful in indicating “effective climate sensitivity” is unsupported. Besides deleting panel f, the discussion of it and of “effective climate sensitivity” would also, of course, need to be eliminated.

Reply: We appreciate the reviewer bringing up this issue. We think the reviewer raises a reasonable

point, but after thinking this over and discussing it extensively between the authors, we have ultimately decided to retain the inclusion of λ_{net}^{eff} and EffCS, because the EffCS is widely used so we see this as an important point of contact with other literature. However, we have added some discussion of the shortcomings of this framework, along the lines of what the reviewer brings up, including the following (lines 170-173):

Note that λ_{net}^{eff} remains negative for all climates, in contrast with λ_{net}^{diff} , which illustrates how the EffCS and λ_{net}^{eff} framework can give potentially misleading results about the stability of the underlying climate state because it is based on anomalies from the PI climate.

The second suggestion involves the claim in the concluding section that the results are important because they indicate that reliance on paleoclimate estimates of climate sensitivity (for temperatures 2K cooler than pre-industrial) cannot constrain current model estimates of climate sensitivity. More discussion and qualification of this conclusion are needed. In particular, the statement that “net feedback (and climate stability) change relatively rapidly for temperatures more than 2K colder than the PI” isn’t supported by figure 2, which by eye seems to indicate the climate sensitivity remains within about 10% of its present value even for climates as much as 10K cooler than present. Since the range of climate sensitivities estimated by models is much larger than this, I think the paleoclimate data may still be of substantial value. The authors should therefore be more circumspect in their discussion of the implications (and practical significance) of their results.

Reply: We agree that the statement in the original submission that values “change relatively rapidly for temperatures more than 2K colder than the PI” was poorly chosen, and we have removed this sentence from the revised manuscript.

We have also revised the discussion in an attempt to be to be more circumspect and to emphasize the value of paleoclimate data for constraining ECS, including (lines 236-249):

These results show that the value of α adopted by Sherwood et al. (2020) for the change in λ_{net} at the LGM is much larger than in the CESM2 results, because the “U” shape in Figs. 2d,e causes the feedback at 5K of cooling to be similar to the feedback at the PI. If we were to repeat the Sherwood et al. (2020) analysis using the value of α that we find here for the difference between the LGM and PI feedbacks, our lower value of α would imply a lower modern-day climate sensitivity than Sherwood et al. (2020) found, which amounts to a stronger constraint on the upper bound of the EffCS than they report. This is because the CESM2 results suggest that the LGM may be a more direct analogue to current warming than previously assumed, since the feedbacks are relatively similar. In other words, Sherwood et al. (2020) took λ_{net} to be more negative at the LGM than the modern value, whereas we find that the feedbacks are similar. So a given paleo estimate of the LGM value of λ_{net} implies a similar modern feedback value according to our results, whereas the analysis of Sherwood et al. (2020) would take it to imply a less negative modern feedback and hence a more sensitive modern climate.

Besides the above two suggestions, I have the following comments and suggestions:

Lines 76-86: In the discussion of “ice cover” and its changes, you should mention here that glacial ice amounts were prescribed and held fixed.

Reply: We have revised this text to mention that it is prescribed, explaining that the ice area “includes sea ice, snow cover on land, and prescribed glacial ice cover” (lines 79-80). Note that in the following paragraph we explain that glacial ice “is a specified surface type in CESM2 with an area that does not evolve during the simulations” (line 94-95).

Line 118: You need to indicate how “effective feedback” is estimated for PI conditions (and plotted on fig. 2e) since it is undefined when ΔT is 0. I guess you must interpolate, but you should mention this.

Reply: We have clarified this in the revised manuscript (lines 120-122):

Eq. 2 is calculated from F_{net} after applying a polynomial smoothing. Note that this allows λ_{net}^{eff} to vary smoothly even in the limit $\Delta T \rightarrow 0$, as described in SI Sec. S3 and shown in Fig. S2.

We have also revised Sec. S3 to include (lines 853-857):

For the effective feedback parameters, we smooth each radiative response time series (F_{net} or F_i) using a least-squares fit to a 12th-order polynomial in $(T - T_0)$ that is constrained to go through (T_0, F_0) , where T_0 and F_0 are the surface temperature and radiative response (F_{net} or F_i) averaged over years 480-499 of the PI simulation. This allows the ratio in Eq. 2 to be smooth even in the limit $T \rightarrow T_0$.

Eq. 4: a plus sign is missing between the first two terms on the RHS.

Reply: We thank the reviewer for catching this typo! We have corrected it in the revised manuscript.

Line 219: replace “atmospheric temperature” with “atmospheric lapse rate”.

Reply: We have made this change in the revised manuscript (lines 279-280).

Line 237: indicates that “cloud and lapse-rate feedbacks” are primarily responsible for the decreased stability as climate warms. It looks to me like the contribution from the residual might be as significant as lapse-rate/water vapor feedback, and this should probably be noted.

Reply: I inadvertently overwrote a variable in the code to make the final version of Fig. 3, which caused the wrong quantity to be plotted for the clear-sky residual in the bottom row of Fig. 3, and I somehow failed to notice that the final version of the figure had this error. I have corrected it in the revised manuscript, which shows a considerably smaller clear-sky residual in Fig. 3.

Line 321: I don’t think subsections are labeled in the SI, so replace “Sec. 5.2” with “Sec. 5”.

Reply: We thank the reviewer for catching this! We have corrected it in the revised manuscript.

Lines 404-416: Yes, the Hahn et al. (2021) results seem consistent with the present study, but I’m not sure this tells us much about whether the kernels can be held fixed, independent of climate state.

And the fact that the clear-sky residuals are not too large also isn't informative in this regard. How do you know that the kernels don't strongly depend on climate state (especially as climate becomes substantially different from the present one?)

Reply: The reviewer raises fair points. We agree that our analysis does not quantify the full-sky residual (since it's wrapped into the cloud feedback). Regarding the Hahn et al. (2021) result, we agree that this does not assess the accuracy of the kernel analysis, as the original submission implied. We have revised the manuscript to clarify that this result provides an assessment of the extent to which equilibration matters, moving the paragraph to our discussion of equilibration (lines 510-520) and revising it accordingly.

Line 420: replace "Cooling" with "cooling".

Reply: We have replaced this with "in colder climates" (line 481)

Line 441: replace "was" with "were".

Reply: We appreciate this correction and have made the suggested change in the revised manuscript.

Line 451: replace "lead" with "led" and replace "surface temperature" with "transient surface temperature changes".

Reply: We appreciate this correction and suggestion and have made both of these changes in the revised manuscript.

Reviewers' Comments:

Reviewer #1:

Remarks to the Author:

The authors have revised the manuscript according to the reviewers' comments. I enjoyed reading the revised manuscript very much.

One more minor comment for the authors to consider: One advantage of the authors' approach is the simplicity and effectiveness of the experiment, i.e., running two very simple sensitivity simulations of 200–500 years each that can continuously cover a temperature range of 55K and the associated fast feedback processes. I think this strength could be better stressed, e.g., in the last paragraph of the manuscript.

Reviewer #2:

Remarks to the Author:

I have reevaluated the manuscript and am pleased to note that the authors have effectively addressed all of my concerns. While I maintain reservations about certain quantitative aspects of the methodology, these concerns have been explicitly discussed in the revised manuscript, enhancing overall clarity.

Given the significant findings, particularly the identification of a U-shaped temperature dependence, I believe the manuscript offers valuable contributions to the scientific community. Therefore, based on the authors' responsiveness to feedback and the noteworthy findings presented, I recommend accepting the paper for publication.

Reviewer #3:

Remarks to the Author:

The authors have adequately addressed most of my concerns, but on rereading the article I stumbled across what might be a real problem I missed before:

According to Smith et al. (<https://doi.org/10.5194/acp-20-9591-2020>), the CMIP6 atmospheric model fast adjustments (stratospheric primarily) increase radiative forcing by about 50% over the instantaneous radiative forcing. Since it is this adjusted forcing that should appear in eq. 1, the subsequent analysis needs to take that into account. Under "Caveats" the potential impact of accounting for stratospheric and tropospheric adjustments is considered, but it is difficult for me to understand (given the above result) why the adjustment is only about 3%. There needs to be some discussion about this. I would note that the lambda curves that are shown in fig. 2 would be substantially altered if in eq. 1, an effective radiative forcing were 50% larger than the instantaneous radiative forcing. It shouldn't be too difficult to consider how that might affect the conclusions.

I also suggest the following:

Line 133: replace "both" with "each".

Line 80: replace "prescribed glacial ice cover" with "prescribed fixed glacial ice cover".

Lines 141-153: I still think these two paragraphs and panel f of figure 2 are of no value. Why is EffCS of interest? To my knowledge, this is the first time EffCS has been defined, at least in this way. It clearly has no relationship whatsoever to equilibrium climate sensitivity because λ_{2x} in the definition of ECS is not the same as $\lambda^{\text{eff_net}}$. ECS allows us to estimate

what the equilibrium temperature will be to a doubling of CO₂ assuming λ is independent of climate state. I have no idea how to interpret EffCS. It's simply inversely proportional to $\lambda^{\text{eff_net}}$, but as a temperature it seems meaningless to me. If you keep these two paragraphs, perhaps you could say why they can help us understand the climate system. Why is EffCS a useful measure, given that we have $\lambda^{\text{eff_net}}$ to gauge climate sensitivity.

Line 279: The Planck feedback reflects not only changes in surface temperature, but also atmospheric temperature, usually assumed to undergo the same change as the surface. This sentence should be revised to correctly characterize the Planck feedback.

Line 284: replace "are shown" with "is shown".

Responses to Reviewer #1

We thank the reviewer for this comment. We have addressed it as follows in the revised manuscript (reviewer's comment in italics):

The authors have revised the manuscript according to the reviewers' comments. I enjoyed reading the revised manuscript very much.

One more minor comment for the authors to consider: One advantage of the authors' approach is the simplicity and effectiveness of the experiment, i.e., running two very simple sensitivity simulations of 200-500 years each that can continuously cover a temperature range of 55K and the associated fast feedback processes. I think this strength could be better stressed, e.g., in the last paragraph of the manuscript.

Reply: We appreciate this suggestion, which we have adopted by adding the following to the penultimate paragraph of the manuscript (lines 563-565): "It is noteworthy that the 279-year and 514-year CO₂ ramping simulations generated for this analysis could be fairly straightforwardly repeated with a different ESM."

Responses to Reviewer #2

We are glad the reviewer finds that we have sufficiently addressed their comments (reviewer's comment in italics):

I have reevaluated the manuscript and am pleased to note that the authors have effectively addressed all of my concerns. While I maintain reservations about certain quantitative aspects of the methodology, these concerns have been explicitly discussed in the revised manuscript, enhancing overall clarity.

Given the significant findings, particularly the identification of a U-shaped temperature dependence, I believe the manuscript offers valuable contributions to the scientific community. Therefore, based on the authors' responsiveness to feedback and the noteworthy findings presented, I recommend accepting the paper for publication.

Responses to Reviewer #3

We thank the reviewer for these comments. We have addressed them as follows in the revised manuscript (reviewer's comments in italics):

The authors have adequately addressed most of my concerns, but on rereading the article I stumbled across what might be a real problem I missed before:

According to Smith et al. (<https://doi.org/10.5194/acp-20-9591-2020>), the CMIP6 atmospheric model fast adjustments (stratospheric primarily) increase radiative forcing by about 50% over the instantaneous radiative forcing. Since it is this adjusted forcing that should appear in eq. 1, the subsequent analysis needs to take that into account. Under “Caveats” the potential impact of accounting for stratospheric and tropospheric adjustments is considered, but it is difficult for me to understand (given the above result) why the adjustment is only about 3%. There needs to be some discussion about this. I would note that the lambda curves that are shown in fig. 2 would be substantially altered if in eq. 1, an effective radiative forcing were 50% larger than the instantaneous radiative forcing. It shouldn't be too difficult to consider how that might affect the conclusions.

Reply: We thank the reviewer for raising this point. We should have been clearer that the close agreement between IRF calculated by radiative transfer code and ERF from CESM2 does not imply that stratospheric and tropospheric adjustments are small. It simply means that CESM2's ERF value (after adjustments) ends up similar to the forcing values we are using, giving us confidence in our calculation. We clarify this interpretation in the Caveats section of the manuscript, where we have added the following (lines 450-456):

The close agreement between the IRF estimate from the radiative transfer code and the ERF estimate from CESM2 may be coincidental given that CESM2, like most ESMs, shows substantial forcing adjustments from rapid changes in atmospheric temperature and cloud cover in response to CO₂ changes (e.g., Smith et al., 2020). However, this agreement gives confidence in the use of the IRF estimate (Fig. 2a) as an approximation to the ERF in CESM2 for our calculations.

I also suggest the following:

Line 133: replace “both” with “each”.

Reply: Agreed, and thanks for the suggestion, which we have adopted.

Line 80: replace “prescribed glacial ice cover” with “prescribed fixed glacial ice cover”.

Reply: We clarified this point by revising the quoted text to “prescribed time-invariant glacial ice cover”.

Lines 141-153: I still think these two paragraphs and panel f of figure 2 are of no value. Why is EffCS of interest? To my knowledge, this is the first time EffCS has been defined, at least in this way. It clearly has no relationship whatsoever to equilibrium climate sensitivity because λ_{2x} in the definition of ECS is not the same as $\lambda_{\text{eff_net}}$. ECS allows us to estimate what the equilibrium temperature will be to a doubling of CO₂ assuming λ is independent of climate

state. I have no idea how to interpret EffCS. It's simply inversely proportional to $\lambda_{\text{eff_net}}$, but as a temperature it seems meaningless to me. If you keep these two paragraphs, perhaps you could say why they can help us understand the climate system. Why is EffCS a useful measure, given that we have $\lambda_{\text{eff_net}}$ to gauge climate sensitivity.

Reply: The effective climate sensitivity (EffCS) is a commonly used metric that allows us to illustrate how the effective radiative feedback relates to the sensitivity of the climate to CO₂ forcing. Our definition of EffCS, as the forcing from CO₂ doubling divided by the effective feedback, exactly follows many previous studies (e.g., Murphy, 1995; Winton et al., 2010; Armour et al., 2013; Sherwood et al., 2020; Mitevski et al., 2021; Andrews et al., 2022). See Rugenstein and Armour (2021) for a discussion of reasons why EffCS can be considered a useful metric.

Line 279: *The Planck feedback reflects not only changes in surface temperature, but also atmospheric temperature, usually assumed to undergo the same change as the surface. This sentence should be revised to correctly characterize the Planck feedback.*

Reply: That is a fair point. We replaced “surface” with “surface and atmospheric column above” in the revised manuscript.

Line 284: *replace “are shown” with “is shown”.*

Reply: We appreciate the reviewer catching this grammar mistake and have corrected it as suggested.

References

- Andrews, T., et al., 2022: On the effect of historical SST patterns on radiative feedback. *J. Geophys. Research-atmospheres*, **127** (18), e2022JD036675, doi:10.1029/2022JD036675.
- Armour, K. C., C. M. Bitz, and G. H. Roe, 2013: Time-varying climate sensitivity from regional feedbacks. *J. Climate*, **26** (13), 4518–4534, doi:10.1175/JCLI-D-12-00544.1.
- Mitevski, I., C. Orbe, R. Chemke, L. Nazarenko, and L. M. Polvani, 2021: Non-monotonic response of the climate system to abrupt CO₂ forcing. *Geophys. Res. Lett.*, **48** (6), e2020GL090861, doi:10.1029/2020GL090861.
- Murphy, J. M., 1995: Transient-response of the Hadley Center coupled ocean-atmosphere model to increasing carbon-dioxide. Part III: Analysis of global-mean response using simple models. *J. Climate*, **8** (3), 496–514, doi:10.1175/1520-0442(1995)008<0496:TROTHC>2.0.CO;2.
- Rugenstein, M. A. A. and K. C. Armour, 2021: Three flavors of radiative feedbacks and their implications for estimating equilibrium climate sensitivity. *Geophys. Res. Lett.*, **48** (15), e2021GL092983, doi:10.1029/2021GL092983.
- Sherwood, S. C., et al., 2020: An assessment of earth's climate sensitivity using multiple lines of evidence. *Rev. Geophysics*, **58** (4), e2019RG000678, doi:10.1029/2019RG000678.
- Winton, M., K. Takahashi, and I. M. Held, 2010: Importance of ocean heat uptake efficacy to transient climate change. *J. Climate*, **23** (9), 2333–2344, doi:10.1175/2009JCLI3139.1.

Reviewers' Comments:

Reviewer #3:

Remarks to the Author:

The authors have adequately addressed the concerns I raised earlier, so I think the article can be published.

I continue to be uncomfortable with how the "transient" response is relied on to infer the dependence of climate sensitivity on climate state and will take the liberty to suggest some obvious follow-up work. Perhaps the authors have already considered this.

It would be interesting to perform an "abrupt quartering" CO₂ experiment to complement the abrupt quadrupling experiment. From these two experiments one could fit results to obtain the appropriate parameters for Geoffroy's 2-layer EBM representation of CESM2 under warming conditions and under cooling conditions. We might expect the parameters to differ since ocean heat uptake would likely have a strong dependence on whether ocean stratification was becoming increasingly or decreasingly stratified. With these parameters, figures could be constructed similar to S4 in the supplemental material. The red and purple curves shown there would show a discontinuity at the initial temperature (control climate temperature), with the curve to the left of it dipping more steeply and to a more negative value. When the composite results of the heating and cooling run are smoothed, one would get a curve that reaches a minimum at a temperature cooler than the control climate, and the sharp "V" point would be rounded into a "U" shape, in a position that might be similar to the blue curve in fig. S4. This would provide a very simple explanation of the results of the CESM2 runs that would not require a strong dependence of feedback strength (and climate sensitivity) on climate state.

The above would not rule out a dependence of feedback strength on climate state but is appealing because it simply explains why the minimum sensitivity is offset from the control climate temperature and shows a general increase (that slows with change) over other temperatures. I find this so appealing that I worry that in fig. S4 the parameters used to construct the red and purple curves may be incorrect. If they are indeed correct, then the contribution of the "transient" impact on the estimate of λ -eff seems to be much too small to explain but a small fraction of the overall behavior. Still, if this were my research, I would double check whether really accurate values for the 2-layer EBM model might give something close to the behavior of CESM2 under the assumption that the parameters are constants but may differ between a warming and cooling climate.

If the transient behavior is responsible for a non-negligible portion of the apparent dependence of λ -eff on climate state, one could use the warming/cooling EBM model runs to subtract these contributions from the total λ -eff obtained from the CESM3 simulations. This would isolate the contribution to variations in climate sensitivity due to variations in the climate state alone.

One final detail: in the caption to figure 4 (in the main text), two references are made to Fig. 2, I think these should refer to Fig. 3.

Responses to Reviewer #3

We appreciate the reviewer taking the time to further reflect on our manuscript. Our responses are below (reviewer’s comment in italics):

The authors have adequately addressed the concerns I raised earlier, so I think the article can be published.

Reply: We are glad the reviewer finds that we have sufficiently addressed their comments.

I continue to be uncomfortable with how the “transient” response is relied on to infer the dependence of climate sensitivity on climate state and will take the liberty to suggest some obvious follow-up work. Perhaps the authors have already considered this.

It would be interesting to perform an “abrupt quartering” CO₂ experiment to complement the abrupt quadrupling experiment. From these two experiments one could fit results to obtain the appropriate parameters for Geoffroy’s 2-layer EBM representation of CESM2 under warming conditions and under cooling conditions. We might expect the parameters to differ since ocean heat uptake would likely have a strong dependence on whether ocean stratification was becoming increasingly or decreasingly stratified. With these parameters, figures could be constructed similar to S4 in the supplemental material. The red and purple curves shown there would show a discontinuity at the initial temperature (control climate temperature), with the curve to the left of it dipping more steeply and to a more negative value.

When the composite results of the heating and cooling run are smoothed, one would get a curve that reaches a minimum at a temperature cooler than the control climate, and the sharp “V” point would be rounded into a “U” shape, in a position that might be similar to the blue curve in fig. S4. This would provide a very simple explanation of the results of the CESM2 runs that would not require a strong dependence of feedback strength (and climate sensitivity) on climate state.

Reply: We agree that the ocean heat uptake could depend on whether the climate is warming or cooling (as investigated, for example, in Stouffer, 2004: “Time Scales of Climate Response”, J. Climate 17, 209-217), and that this can be modeled by varying the parameters in the two-layer model results shown in our Fig. S4. To illustrate this, we have repeated the result, using the large value of ϵ as in the purple curve in our Fig. S4b, but this time doubling the value of γ for the cooling simulation (left half of figure). The result is shown below (purple curve). Next, we smoothed this two-layer model result using the same procedure that we used for the CESM results, which is also included below (green curve).

The reviewer's hypothesis is that it is the smoothing of the results that causes the minimum of the "U" to be offset to a temperature colder than the starting value. And consistent with this, the green curve in the figure above does have a minimum temperature that is 1.4°C colder than the starting temperature (compared with 4.8°C colder in the CESM2 results).

Nonetheless, it can readily be seen from the results in the manuscript that this hypothesis does not explain the CESM2 results. Specifically, Fig. S2b of the manuscript shows the raw unsmoothed CESM2 results (blue dots). The blue dots representing the cold climates nearest to the reference climate (blue dots immediately to left of vertical dashed line) do not indicate increasingly negative values (as in the purple curve above), instead showing the beginnings of an upturn toward less negative values.

In other words, the reviewer suggests an interesting alternative hypothesis for our results which would suggest that the key features we focus on are actually an artifact of the smoothing technique that we employed, but the figure we include to evaluate our smoothing technique (Fig. S2b) shows that our results are not actually consistent with the reviewer's hypothesis.

The above would not rule out a dependence of feedback strength on climate state but is appealing because it simply explains why the minimum sensitivity is offset from the control climate temperature and shows a general increase (that slows with change) over other temperatures. I find this so appealing that I worry that in fig. S4 the parameters used to construct the red and purple curves may be incorrect. If they are indeed correct, then the contribution of the "transient" impact on the estimate of lambda-eff seems to be much too small to explain but a small fraction of the overall behavior. Still, if this were my research, I would double check whether really accurate values for the 2-layer EBM model might give something close to the behavior of CESM2 under the assumption that the parameters are constants but may differ between a warming and cooling climate.

If the transient behavior is responsible for a non-negligible portion of the apparent dependence of lambda-eff on climate state, one could use the warming/cooling EBM model runs to subtract these contributions from the total lambda-eff obtained from the CESM3 simulations. This would isolate the contribution to variations in climate sensitivity due to variations in the climate state alone.

Reply: We appreciate the reviewer raising this alternative hypothesis, which we have considered as described above.

One final detail: in the caption to figure 4 (in the main text), two references are made to Fig. 2, I think these should refer to Fig. 3.

Reply: We appreciate the reviewer catching this typo! We have corrected it in the revised manuscript.